# Estimating the fate of oxygen ion outflow from the high altitude cusp

Patrik Krcelic[1,2], Stein Haaland[1,3], Lukas Maes[1], Rikard Slapak[4], and Audrey Schillings[5,6]

[1]Max-Planck Institute for Solar Systems Research, Göttingen, Germany
[2]Department of Geophysics, Faculty of science, University of Zagreb, Croatia
[3]Birkeland Centre for Space Science, Department of Physics and Technology, University of Bergen, Norway
[4]EISCAT Scientific Association, Kiruna, Sweden
[5]Swedish Institute for Space Physics, Kiruna, Sweden
[6]Division of Space Technology, Lulea University of Technology, Kiruna, Sweden

**Correspondence:** Patrik Krcelic (patrik.krcelic@gmail.com)

**Abstract.** We have investigated the oxygen escape-to-capture ratio from the high altitude cusp regions for various geomagnetic activity levels by combining EDI and CODIF measurements from the Cluster spacecraft. Using a magnetic field model, we traced the observed oxygen ions to one of three regions: plasma sheet, solar wind beyond distant X-line or dayside magnetosheath. Our results indicate that 69 % of high altitude oxygen escapes the magnetosphere, from which most escape beyond the distant X-line (50% of total oxygen flux). Convection of oxygen to the plasma sheet shows a strong dependence on geomagnetic activity. We used the Dst index as a proxy for geomagnetic storms and separated data into quiet conditions ($Dst > 0$ nT), moderate conditions ($0 > Dst > -20$ nT), and active conditions ($Dst < -20$ nT). For quiet magnetospheric conditions we found increased escape due to low convection. For active magnetospheric conditions we found an increase in both parallel velocities and convection velocities, but the increase in convection velocities is higher, and thus most of oxygen gets convected into the plasma sheet (73 %). The convected oxygen ions reach the plasma sheet in the distant tail, mostly beyond $50$ $R_E$.

*Copyright statement.* TEXT

## 1 Introduction

The Earth's magnetosphere is populated with plasma of two different origins: the solar wind and the terrestrial ionosphere. Plasma of terrestrial origin constitutes a considerable part of the total plasma in magnetosphere (Chappell et al., 1987, 2000; Yau and André, 1997; Moore and Horwitz, 2007), and have an important impact on the magnetosphere in general (e.g, Glocer et al., 2009). Lighter ions ($H^+$, $He^+$) in the magnetic lobes mainly originate from the polar cap regions (Axford, 1968; Laakso and Grard, 2002; Kitamura et al., 2011), auroral regions (Yau et al., 1985), and cusp regions (Lockwood et al., 1985). The dominant source region of light ions in the lobes is polar cap. In the cusps, ions typically escape with much higher velocities, but due to the smaller area of the cusp, the total outflow from the cusp is less than from polar cap. Heavier ions ($O^+$) need higher energies ($\geq 10$ eV) to overcome Earth's gravity, and mainly escape from the cusps (Lockwood et al., 1985).

The magnetospheric cusps are narrow regions of open field lines, magnetically connected to the magnetosheath and the solar wind. As a result, the heating in the cusps is higher than in the polar caps. The interaction between the magnetosheath and the magnetosphere leads to a perpendicular energization of ions. Due to strong magnetic gradients in the cusp regions, mirror forces can effectively transform perpendicular energy into parallel energy. The field aligned acceleration from the mirror force becomes sufficient to overcome the gravitational potential for hydrogen and oxygen ions (Nilsson et al., 1996; Ogawa et al., 2003; Kistler et al., 2010). As the main driver of cusp outflow, ion transverse heating has been analyzed in detail in (e.g., Andre et al., 1990; Norqvist et al., 1996; Bouhram et al., 2003; Waara et al., 2011; Slapak et al., 2011).

The fate of escaping oxygen ions is determined by the ratio between their parallel velocity (along the magnetic field) and the convection velocity (perpendicular to the magnetic field). For given solar wind conditions, both convection velocity and parallel velocity increase with radial distance inside the magnetosphere. The convection velocity scales with the inverse of magnetic field magnitude, whereas the parallel velocity increases due to the combined effect of the mirror force and the centrifugal force.

Engwall et al. (2009) measured cold ions ($< 100$ eV, mostly $H^+$) in the lobe regions and calculated typical values for lobe plasma properties (velocity, density, acceleration, etc.). As estimated by Haaland et al. (2012), most of the $H^+$ ions return to the magnetosphere. The fate of oxygen ions is not fully understood. Seki et al. (2001) concluded that over 90 % of $O^+$ return back to magnetosphere. However, this statement was challenged by Nilsson (2011), claiming that the Seki et al. (2001) study underestimated the outflowing energies of the $O^+$ ions. Seki et al. (2001) used $O^+$ energies lower than 1 keV, while Nilsson (2011) measured the energies in the range $1 - 8$ keV at high altitudes ($> 6$ $R_E$). Ebihara et al. (2006) traced $O^+$ ions and stated that most of them end up feeding the ring current. Their research included oxygen ions with low initial energies $<200$ eV. Slapak and Nilsson (2018) looked for the total oxygen ion outflow from the ionosphere to the magnetosphere and concluded that there are no hidden populations of the oxygen ions. In their paper, oxygen ions originating from the cusps either exit the magnetosphere into the magnetosheath or are bound to the open field lines at $X_{GSM} \approx -20$ $R_E$. Liao et al. (2010) presented a statistical cusp oxygen outflow study and come to similar conclusion; ions originating from the cusps mostly end up on open field lines at $X_{GSM} \approx -20$ $R_E$ distances.

A significant part of the acceleration along the magnetic field lines in the cusps is due to centrifugal acceleration (Cladis, 1986; Nilsson et al., 2008, 2010), and thus convection plays a considerable role. Other acceleration processes also take place in the cups and will be further discussed in section 3.

Slapak et al. (2017) used the Composition and Distribution Function (CODIF) ion spectrometer onboard Cluster to get in-situ measurements of $O^+$ and $H^+$ in the cusp and plasma mantle regions. The plasma mantle is a boundary region of the magnetic lobes, neighboring the tailward cusp. They concluded that most of the high altitude oxygen ion outflow is transported to the solar wind beyond the distant X-line or to the dayside magnetosphere. Slapak et al. (2017) did not investigate the role of convection in detail, so in this paper, we further investigate the role of convection in oxygen outflow by combining Electron

Drift Instrument (EDI) and CODIF data. We are trying to answer the question: What fraction of the high altitude cusp oxygen outflow returns to the magnetosphere?

This paper is organized as follows: In section 2 we discuss the key Cluster instruments used and give a short overview of the data sets. The method we use is discussed in detail in section 3, along with its assumptions and shortcomings. In section 4 we present the results for different geomagnetic conditions. Section 5 discusses the results, and a summary and conclusions are given in section 6.

## 2 Data

The Cluster mission consists of four identical spacecraft flying in a tetrahedron-like formation (Escoubet et al., 2001). Cluster has a polar orbit with a period of around 57 hours. Although some modifications in the orbit have been made during the mission, the data used in this paper are mostly from orbits with perigee around $4$ $R_E$ and apogee around $19$ $R_E$. Initially Cluster had its apogee in a near ecliptic plane, but it slowly moved southward due to precession.

Since there are not much simultaneous EDI and CODIF measurements, we combine the two datasets, using EDI and CODIF data taken under similar geomagnetic conditions and in same region in space, but not necessarily simultaneously.

### 2.1 Cluster EDI data

Convection measurements used in this study are obtained from the EDI onboard Cluster. This instrument operates by injecting an electron beam into the ambient magnetic field, and detecting the same beam after one or multiple gyrations. Due to the electron cycloidal motion, the electron beam can only be detected if fired in a unique direction determined by the drift vector. The full velocity vector is calculated from either the direction of the beams (via triangulation, usually for small drift velocities) or from the difference in the time-of-flight of the electrons (usually for bigger drift velocities). The emitted electron beams have energies of $1$ keV (rarely $0.5$ keV) and are modulated with a pseudo-signal in order to be distinguished from ambient electrons. EDI gives precise full 3D coverage, unlike the double probe instrument EFW (Electric-field wave instrument), which gives the E-field in the spin plane (Gustafsson et al., 1997; Pedersen et al., 1998). EDI measurements are also not affected by wake effects nor spacecraft charging, which may affect the EFW instrument and plasma instruments. The accuracy of EDI is not affected by low plasma densities, and actually works better if the plasma density is low. EDI, however, does not provide continuous data, and the data return is reduced in low magnetic field environments ($<20$ nT), or if the ambient magnetic field is too variable. EDI will also have reduced data return in regions of high $\approx 1$ keV background electron flux. Since EDI is an active experiment it can interfere with wave measurements on Cluster, and therefore operates on a negotiated duty-cycle. More information about EDI can be found in Paschmann et al. (1997, 2001); Quinn et al. (2001).

The data set used in this study is from January 2002 until April 2004 for Cluster 2 (C2), from January 2002 until December 2010 for Cluster 1 (C1), and from January 2002 until December 2016 for Cluster 3 (C3). We have used 1-minute EDI data, calculated as the averages (medians) from the EDI spin resolution data set ($\approx 4$ s resolution).

## 2.2 EDI data coverage

In this study we are primarily interested in convection in the cusps. In order to distinguish the cusps from the polar caps the Tsyganenko and Stern T96 magnetic field model (Tsyganenko and Stern, 1996) was used. The reason we chose to use the older model is because we use a statistical approach with over 10 years of data. On these time scales, the newer models (e.g., Tsyganenko, 2002; Tsyganenko and Sitnov, 2005) and older magnetic models do not differ much in the regions relevant for this study.

We identify the cusp regions using the T96 model: The cusps have open field lines which stretch beyond magnetopause. (Since the T96 model is only valid inside the magnetosphere, field lines outside of the magnetosphere are represented as parallel with the IMF.) An example is given in the left panel of Figure 1; cusp field lines are represented in red. We also include plasma mantle data in order to compare our results with Slapak et al. (2017). The plasma mantle, in our study, is chosen as the neighboring regions of the cusp based on the T96 model. The average cusp latitudinal extent in ionosphere is around $4°$

(Newell and Meng, 1987; Burch, 1973). Using model for data selection may not be the most accurate method, but measurements are certainly not in the solar wind (EDI does not work in solar wind). Measurements may be in polar caps in some cases, but we are working with grand averages. The convection velocity obtained form the cusps would not be very different from the one in the polar caps (unlike the parallel velocities which are strongly modulated by heating). Also, the convection velocities obtained from EDI seem sensible, and are what we would expect in the cusps (100-1000 m/s in ionosphere).

   We traced field lines from regions adjacent to the above determined cusps to the ionosphere. If the tracing landed within $2°$ poleward of the cusp, we characterized them as plasma mantle data. The schematic representation is shown in figure 1. The left panel shows the boundary cusp field lines (red) and boundary plasma mantle field line (blue) in the $XZ_{GSM}$ plane.

The right panel depicts cusp (red) and plasma mantle (blue) areas in the ionosphere. For this representation we have assumed longitudinal symmetry of the ionospheric cusps.

   Using the TS96 model to extract 1-minute cusp and plasma mantle measurements, the total number of EDI measurements is 1130 hours (448 hours are from the cusps), whereof 478 (163 from cusps) hours of data are from the northern hemisphere, and 652 (285 from cusps) hours are from the southern hemisphere. The larger number of measurements from the southern

hemisphere is a consequence of the Cluster orbit precession. We have more EDI observations from the plasma mantle than from the cusp, since the variable cusp magnetic field reduces the number of good quality EDI measurements ("good quality" label is given in Cluster Science Archive according to the series of criteria explained in EDI user guide (Georgescu et al., 2010)).

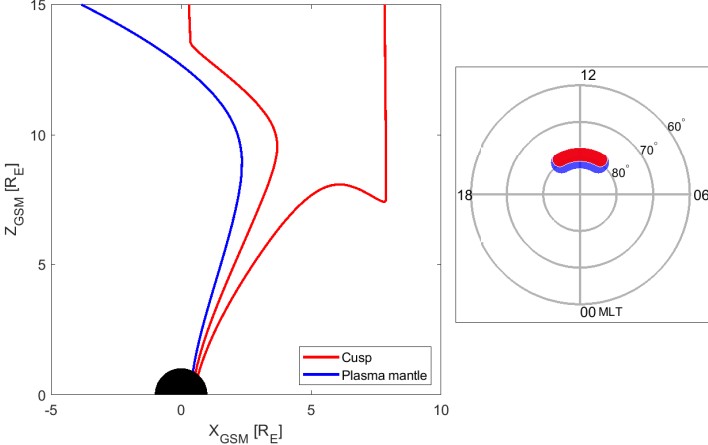

**Figure 1.** Schematic representation of the cusp and plasma mantle regions determined from the T96 model. The left panel depicts boundary field lines in the $XZ_{GSM}$ plane. The right panel depicts schematic (symmetric) areas of the cusp and plasma mantle in the polar cap. The cusp is represented with red, and plasma mantle with blue.

The right panel of figure 2 shows the total distribution of all EDI measurements used. The data are shown in cylindrical GSM coordinate system ($R_{cyl} = \sqrt{Y_{GSM}^2 + Z_{GSM}^2}$), and projected into the northern hemisphere. Here we ignored any north-south asymmetries, and used only data with $R > 6$ $R_E$. The color bar indicates the number of one-minute data in each $1 \times 1$ $R_E$ bin. At least 3 minutes of data in each bin was required. The black line represents the average theoretical magnetopause position as in Shue et al. (1998) with input values of $B_z = -1$ nT and $P_{DYN} = 2$ nPa.

## 2.3 Cluster CODIF Data

In order to measure parallel velocities and ion fluxes, the CODIF spectrometers (Rème et al., 1997) onboard the Cluster spacecraft were used. We use the same data set as used in Slapak et al. (2017) in which plasma mantle data were obtained. A more detailed description of the dataset is given in Slapak et al. (2017), but for convenience we repeat some of the information.

The plasma mantel data set was made using CODIF data from 2001 to 2005. Separating O$^+$ CODIF data in the plasma mantle from the magnetosheath and the polar cap was done using a few criteria. First, the inner magnetosphere was removed by using only data where $R_{GSM} = \sqrt{Y_{GSM}^2 + Z_{GSM}^2} > 6$ $R_E$. In order to exclude polar cap data, the plasma $\beta$ was used (derived from combined O$^+$ and H$^+$ CODIF data). Typical values of plasma $\beta$ in the polar caps are below $0.01$, and in plasma mantle and magnetosheath is above $0.1$. Only data with $\beta > 0.1$, are used. For separation between plasma sheet and plasma mantle data, Slapak et al. (2017) used H$^+$ CODIF data. They noticed two clearly distinct peaks in H$^+$ temperature for data with $\beta > 0.1$. They decided on a H$^+$ ion cut temperature of $1750$ eV to separate the two populations. The two populations had different values of densities as well. One population had higher temperatures and lower densities as expected in the plasma sheet, while the other population had lower temperatures and higher density as expected in the plasma mantle. O$^+$ shows similar fea-

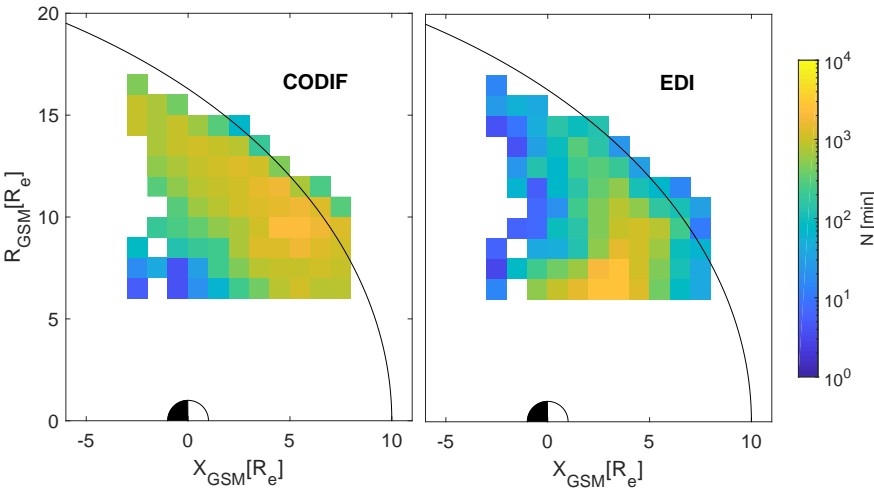

**Figure 2.** Coverage of CODIF and EDI data projected into northern hemisphere. The data are represented in cylindrical coordinate system, where $R_{GSM} = \sqrt{Y_{GSM}^2 + Z_{GSM}^2}$. The color bar indicates number of one-minute measurements in each $1 \times 1$ $R_E$ bin. Left panel depicts CODIF coverage, while right panel depicts EDI coverage.

tures between the two populations. $O^+$ densities in both populations are 1 order of magnitude lower than $H^+$ densities, which is expected, and the plasma mantle population has a wider temperature range. Still, the two populations are easily distinguishable, and only data with $T_\perp < 1750$ eV is used. To separate magnetosheath data from plasma mantle data, Slapak et al. (2017) visually inspected $O^+$ spectrograms. Magnetosheath is a region usually characterised with more fluctuant magnetic field than

inside of magnetosphere. It is also characterised by strong $H^+$ fluxes, which sometimes contaminate the $O^+$ mass channel. In order to get parallel velocities (along the field lines) we used the scalar product of oxygen ion velocities and magnetic field direction: $v_\parallel = \boldsymbol{v_O} \cdot \boldsymbol{b}$ , where $\boldsymbol{V_O}$ is a oxygen ion velocity measured with CODIF and $\boldsymbol{b}$ is the direction of the magnetic field. We have then easily calculated the perpendicular velocity as: $\boldsymbol{v_\perp} = \boldsymbol{v_O} - v_\parallel \boldsymbol{b}$. The perpendicular $O^+$ velocities are comparable to the parallel velocities. We believe that the $\boldsymbol{v_\perp}$ values are too high and cannot be explained with the convection. As of now

we do not know how to explain the perpendicular velocities and choose to ignore them in this study, and use more accurate EDI measurements instead.

    In total we have 1422 hours of CODIF measurements. The distribution of CODIF measurements is shown in the left panel of figure 2. Here we can see the difference in data coverage between the two instruments (EDI and CODIF). The main reason

for this asymmetry are the technical restrictions of the instruments. EDI has fewer measurements closer to the magnetopause because of higher variability of the magnetic field, while CODIF has more measurements closer to the magnetopause due to higher fluxes in this region. In addition to EDI and CODIF Cluster data, we also used solar wind dynamic pressure, Dst and IMF data from the OMNI dataset (King and Papitashvili, 2005). For IMF and solar wind dynamic pressure we used one minute values and for Dst we have implemented a simple linear interpolation in order to get the one minute values.

## 3    Method

The method used is a combination of the ones described in Haaland et al. (2012) and Li et al. (2012). If the outflowing ions can be traced to closed magnetic field lines before they reach the distant X-line at ca $-100\ R_E$ (e.g., Grigorenko et al., 2009; Daly, 1986), we inter that they are captured and returned to the magnetosphere. If they reach the X-line before being convected

to the plasma sheet, the ions will be lost into the solar wind. For the highest energies, some of the ions will escape into the dayside magnetosheath directly before being convected into the plasma mantle. One issue here is the position of the distant X-line, which is not permanent, but can vary with geomagnetic conditions. Since we do not know the exact location of the distant X-line in relation to the geomagnetic conditions, we have decided to use the fixed X-line and coment its effect on the results in the discussion section. Another issue is the forming of the near Earth X-line (around $X_{GSM} = -20\ \mathrm{R_E}$) during active

geomagnetic conditions. At this point we are unable to determine what happens to the ions that are landing two X-lines. The method we use to track the ions along their paths is based on the tracing of the ions along the field line using the TS96 model, and moving the field lines with each time step in order to simulate the convection. We used the CODIF measurements of $v_\parallel$ to move the ions along the field line in each time step and EDI measurements of $v_\perp$ to move the field line accordingly. The method described in Haaland et al. (2012), infers that the capture will depend on the location of the ions in the $YZ_{GSM}$ plane

at $X_{GSM} = -10\ R_E$. In their study the velocities and accelerations were calculated as averages. In Li et al. (2012) ions were traced for each measurement of the parallel and convection velocity. They calculated the acceleration for each tracing step. The direction and magnitude of the convection velocity are given by the following equation:

$$\boldsymbol{v_{i,d}} = |\boldsymbol{v_{0,d}}|\sqrt{\frac{|B_0|}{|B_i|}}\Big(\frac{(\boldsymbol{B_i}\cdot\nabla)\boldsymbol{B_i}}{|(\boldsymbol{B_i}\cdot\nabla)\boldsymbol{B_i}|}\Big), \tag{1}$$

where the subscript $0$ indicates the initial velocity and magnetic field, and $i$ denotes the $i$-th step. In present paper we use a

method similar to that of Haaland et al. (2007) to sample measurements and the method of Li et al. (2012) to trace particles.

Compared to the polar cap, ions escaping from the cusps have a broader energy range $15\ \mathrm{eV}$-$5\ \mathrm{keV}$ (e.g., Bouhram et al., 2004; Lennartsson et al., 2004; Nilsson et al., 2012), so the mirror force and hence the acceleration and parallel velocity will vary correspondingly.

The location of the observations is very important since there is a region of enhanced perpendicular heating in the cusps in the range $8$-$12\ \mathrm{R_E}$ (Arvelius et al., 2005; Nilsson et al., 2006; Waara et al., 2010), which results in higher perpendicular energies and thus higher parallel velocities due to the mirror force. If the outflowing ions are convected across the cusp to the plasma mantle before reaching this perpendicular heating region, they will not be significantly energized and will retain small energies and velocities. On the other hand, if they reach this heating region, they will be accelerated and can either be convected

into the plasma mantle with large energies and velocities, or escape into the dayside magnetosheath before being convected into closed magnetic field lines. In Nilsson et al. (2008), the centrifugal acceleration analysis in the cusp is discussed in some detail. There is significant acceleration between $8$ and $10\ R_E$. The acceleration in that region cannot be described by centrifugal

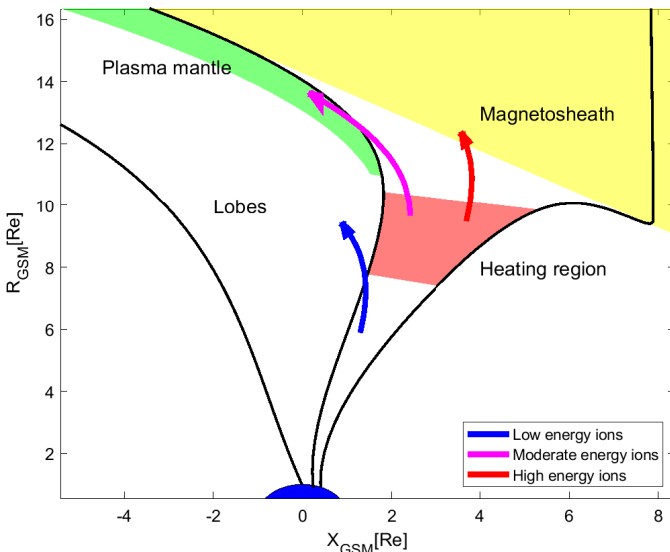

**Figure 3.** Paths of oxygen ions based on their energies. The heating region in the high altitude cusps as well as lobe and magnetosheat regions are included

acceleration alone, and is most likely acceleration caused by wave particle interaction. Figure 3 shows typical transport paths for oxygen ions of low, intermediate and high energies.

Our main assumption is that only centrifugal force accelerates oxygen ions on their path (mirror force acceleration is included in the centrifugal acceleration from Nilsson et al. (2008)). A further assumption is that no other energization takes place along
5 the particle path outside the cusps (e.g. no parallel E-fields or wave-particle acceleration). The gravitational force has no effects on the accelerations for the altitudes considered in our research, and without further energization the mirror force has little effect outside the cusps. We assume steady solar wind conditions during the tracing.

For particle acceleration along the field line we use two values of the centrifugal accelerations; one value for the cusp and a different value for the lobe as in Nilsson et al. (2008, 2010). For cusp acceleration we used the following values:

$$10 \quad a_c = \begin{cases} 12 \text{ ms}^{-2} & \text{if } R < 8 \text{ R}_{\text{E}} \\ 100 \text{ ms}^{-2} & \text{if } 8 < R < 9 \text{ R}_{\text{E}} \\ 70 \text{ ms}^{-2} & \text{if } R > 9 \text{ R}_{\text{E}} \end{cases} \tag{2}$$

For lobe acceleration, $a_l$, we used $a_l/r = 60 \text{ m s}^{-2}\text{R}_{\text{E}}^{-1}$, where the acceleration is scaled with radial distance given in Earth radii. The resulting velocity versus radial distance is shown in figure 4. The red line represents cusp velocities, and the blue line represents lobe velocities.

From the EDI measurements in the cusp regions (based on the TS96 model) we have calculated the average convection
15 velocity scaled to the ionosphere (height where B = $50000$ nT, as in Slapak et al. (2017)). The average cusp convection velocity in the ionosphere is $620 \text{ ms}^{-1}$ in our data set (at $\approx 400$ km altitude). As an average cusp size in the ionosphere

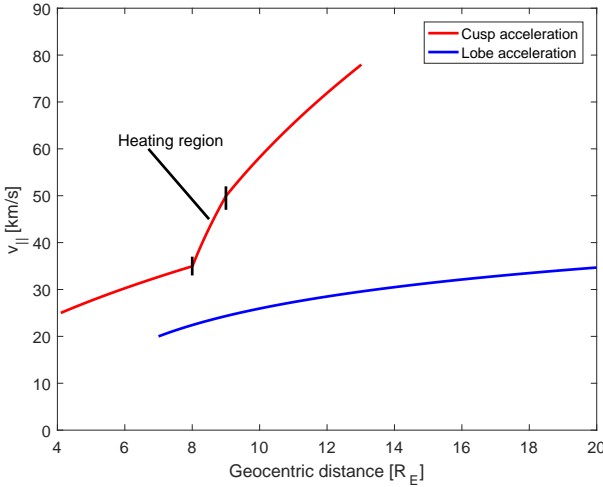

**Figure 4.** Velocity dependence on radial distance in cusps and lobes, using acceleration values given in text. Red line represents cusp velocities and the blue line represents lobe velocities.

we used $4°$ in latitude (Burch, 1973). The average time to convect the most equatorward cusp field line across the cusp, is 11 minutes. Newell and Meng (1987) calculated cusp widths as function of the IMF $B_z$ component. They investigated two case studies of changing IMF direction from northward to southward direction. In first case they had stronger IMF for both southward and northward direction which resulted in $3.5°$ latitudinal extent for northward IMF and $2°$ for southward IMF. In

second case they reported $1.7°$ latitudinal extent of cusps for northward IMF and $0.7°$ for southward IMF. For the latter case, Newell and Meng (1987) concluded that for northward IMF the cusp size decreased due to ongoing nightside reconnection and for southward IMF the cusp size decreased because strong convection rapidly closed the open cusp field lines. In this study we used values from the first case in the Newell and Meng (1987): $3.5°$ for northward IMF and $2°$ for southward IMF. For average IMF conditions we have decided to use $4°$ cusp latitudinal extent as given in Burch (1973). The cusp latitudinal extent, $\Delta\phi$,

and scaling of cusp convection, $v_{SC,\perp}$, to the ionosphere, $v_{i,\perp}$, are illustrated in figure 5.

The starting point of our tracing is the center of each $1 \times 1$ $R_E$ spatial bin shown in figure 6. In order to avoid any dawn-dusk asymmetries we use $Y_{GSM} = 0$ and $Z_{GSM} = R_{cyl}$ as the starting point. The initial convection velocity is given by the measurements in each spatial bin. Convection velocities used are shown in the figure 6. Convection velocities in each bin are calculated as the median of all measured drift magnitudes within a given bin. Average directions are calculated as the median

value of the components of the normalized vectors. In the figure 6, average convection velocities are shown with arrows. The length of the arrow indicates the magnitude of the vector; the scale is given in upper right corner. Colors of the bin represent the bias vector. The bias vector is calculated as magnitude of the mean vector calculated from an ensemble of normalized vector components:

$$|\boldsymbol{B_v}| = |\langle \frac{\boldsymbol{v}}{|\boldsymbol{v}|} \rangle|, \tag{3}$$

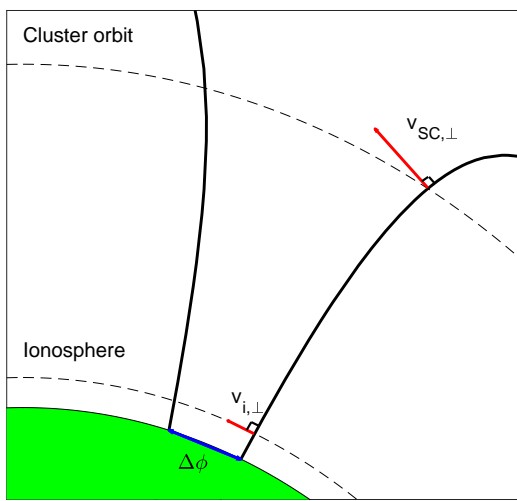

**Figure 5.** Illustration of rescaling convection measurement to ionospheric height. The measured velocity at the spacecraft location, $v_{SC\perp}$ is scaled to ionospheric height $v_{i,\perp}$. $\Delta\phi$ is cusp latitudinal extent at the surface of the Earth. Black lines represent the most sunward and the most tailward cusp field line, respectively.

where, $v$ represents measured velocities and $\langle...\rangle$ denotes mean value. The bias vector is a good estimate of angular spread (see Haaland et al. (2007)). Bias values close to zero value indicate a highly variable vector direction distribution, while values close to unity indicate vectors pointing in a coherent direction. Figure 6 shows that the direction of convection in the cusps is very variable. Bias vector values around $0.8$ indicate an angular spread of around $\pm 45°$. We see that in the cusps the bias vector values are often lower than $0.8$, indicating very variable convection direction. This variability comes from the dynamic nature of the cusps. The cusp position and size are constantly changing due to solar wind conditions ($IMF$, $P_{Dyn}$) as well as temporal variations in tilt angle (daily and seasonal). Therefore, when averaging convection velocities without separation of the magnitude and direction, the average velocity will have a much smaller value, than when averaging only the magnitude.

Since we use a magnetic field model, the initial convection velocity is given by the average(median) of the magnitudes within a bin, and the direction of the convection velocity is calculated using eq. (1). The same equation is used to evaluate convection for further steps. For the parallel velocity we used median values from the CODIF dataset (Slapak et al., 2017) as magnitude, and a direction is given by the magnetic field model. For the subsequent time step we add acceleration. The first 11 minutes we use the cusp acceleration, given in Nilsson et al. (2008), and for the rest of the steps we use lobe acceleration values from Nilsson et al. (2010) - see Equation 2. The distance travelled by a particle within one time step is then the product of the velocity times the time step. We have arbitrarily chosen a time step of one minute. If the particle exits the magnetosphere within the first 11 minutes, we say that it has escaped into the dayside magnetosheath. If the particle ends up on closed field line before reaching the X-line we say it has returned to magnetosphere. If the particle reaches the plasma sheet beyond the

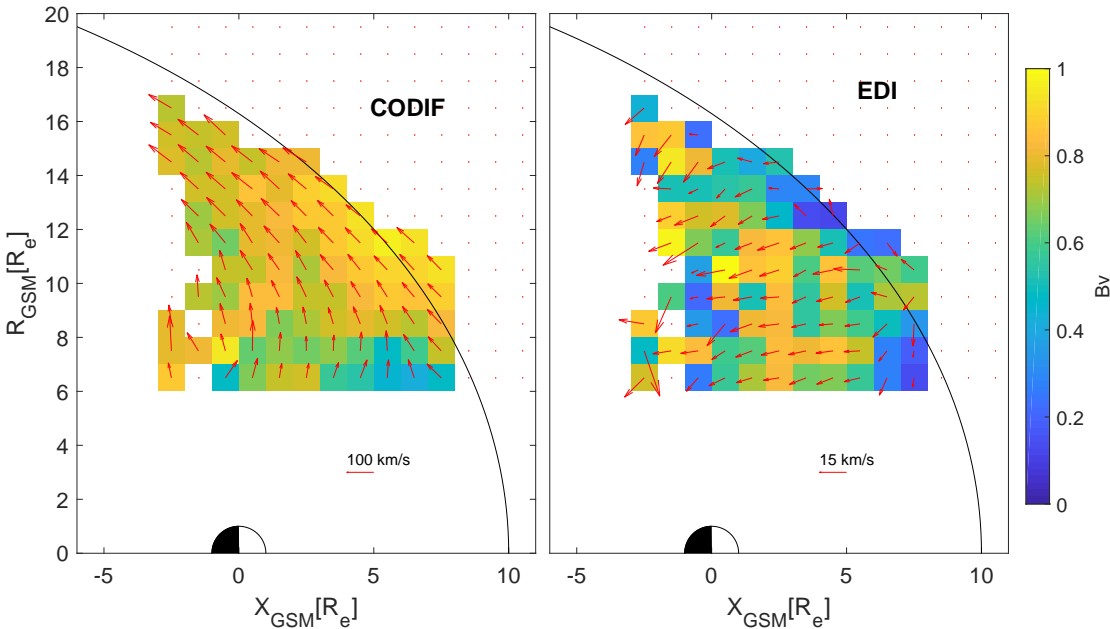

**Figure 6.** Distribution of average parallel and convection velocities in the cusps and plasma mantle regions. Lengths of vectors represent convection velocity in each bin, calculated as the average magnitude of vectors. Colors indicate bias value in each bin, a measure of directional variability. Left panel depicts parallel velocities obtained using CODIF data; right panel depicts convection velocities obtained using EDI data. Vectors are scaled as given in the lower right corner of each panel.

distant X-line, we say it escapes into the solar wind. A drawback with this simple separations of final regions is that they are based on a static model (T96), but this is as good as we can do with the present models.

To estimate the percentages of oxygen outflow which end up in each of the 3 regions; solar wind, magnetosheath, plasma sheet, we use the average measured oxygen flux in each bin. Depending on where each oxygen trace line ends, we add the average flux of that bin to the total flux of the respective region. Figure 7 shows oxygen flux distribution in the measurement bins.

For the time input parameter to initialize the T96 model, we used the time of the equinox at noon for year 2011 (21.03.2011. 12:00:00). We have chosen the equinox because it represents (more or less) a yearly average state of magnetosphere in our dataset. We decided to use the spring equinox since in March the Cluster apogee is in the solar wind, and Cluster passes trough the dayside magnetosheath. Therefore, during spring the equinox we have more measurements than during the autumn equinox. We chose 2011 because it is in the middle between minimum and maximum of the solar cycle.

The rest of the input parameters (Dst, IMF and solar wind pressure) are taken as the median of all values in the respective parameter. Results within a given Dst range are median values of a measurements within that Dst range. Input parameter values used for each condition are shown in table 1. In table 1 we also present the ionospheric cusp latitudinal extents from Newell and Meng (1987) ($\Delta\phi$ in the table). Note that Newell and Meng (1987) correlated cusp width with the IMF Z-

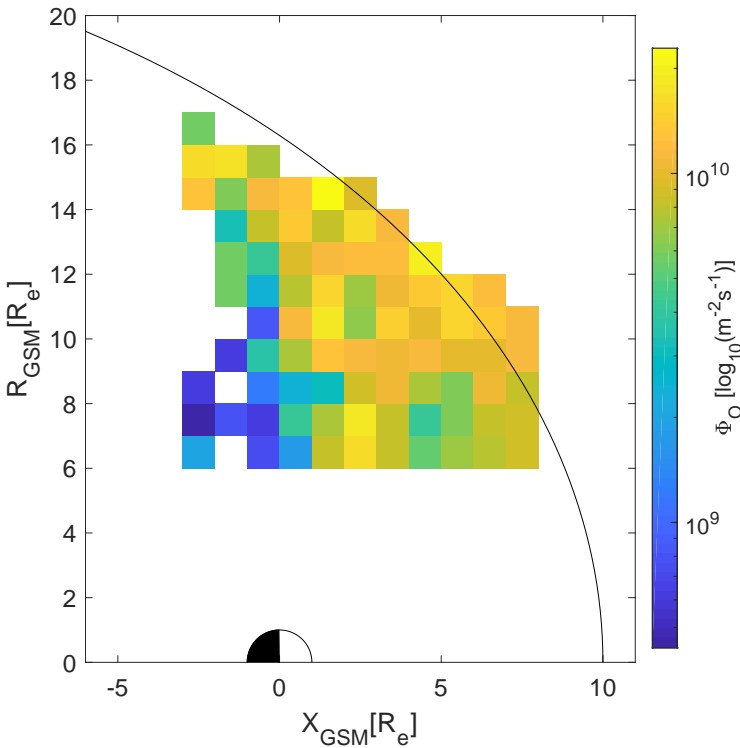

**Figure 7.** Oxygen flux distribution in each measurement bin. Here we only use bins with both parallel and convection velocity data. Colorbar indicate the amount of flux in each bin scaled to ionospheric level (50000 nT)

**Table 1.** Used input parameters in geomagnetic model for different conditions. The first column shows the corresponding average of full data set.

|  | All | $Dst > 0$ | $-20 < Dst < 0$ | $Dst < -20$ |
|---|---|---|---|---|
| $p_{DYN}[\text{nPa}]$ | 1.5 | 1.8 | 1.3 | 1.6 |
| $Dst[\text{nT}]$ | $-17.2$ | 4.7 | $-10.1$ | $-41.6$ |
| $B_{IMF}^{Z}[\text{nT}]$ | $-0.9$ | 0.5 | $-0.5$ | $-2.3$ |
| $B_{IMF}^{Y}[\text{nT}]$ | $-0.1$ | 0.2 | 0 | $-0.6$ |
| $v_{i,c}[\text{ms}^{-1}]$ | 630 | 505 | 616 | 708 |
| $\Delta\phi[°]$ | 4 | 3.5 | 4 | 2 |
| $t_c[\text{min}]$ | $\approx 11$ | $\approx 12$ | $\approx 12$ | $\approx 4$ |

component, while we are using Dst to group the measurements. As seen from table 1, the average IMF conditions for a given Dst range are in good agreement with Newell and Meng (1987). The other parameters in table 1 are the average cusp convection scaled to ionospheric level ($v_{i,c}$), and the maximum cusp convection time ($t_c$).

**Table 2.** Estimated fate of oxygen ions expressed as percentages of outflow flux. $\Phi$ represents the flux, and subscripts $ms$, $sw$ and $ps$ represent magnetosheath, solar wind and plasmasheet respectively. $\sigma_{par}$ represent the standard deviation of the parallel initial velocities.

|  | $\langle v_{par} \rangle$ | $\langle v_{par} \rangle - \sigma_{par}$ | $\langle v_{par} \rangle + \sigma_{par}$ |
|---|---|---|---|
| $\Phi_{ms}$ | 18 % | 15 % | 19 % |
| $\Phi_{sw}$ | 50 % | 37 % | 63 % |
| $\Phi_{ps}$ | 31 % | 48 % | 18 % |

## 4 Results

Figure 8 shows average particle traces for each $1 \times 1$ $R_E$ measurement bin. Colors indicate where the ions will end up. Blue color represents ions returned to the magnetosphere (captured), red color indicate the path of particles passing the X-line (lost), ending up in the solar wind, black color indicate paths of ions transported to the dayside magnetosheath (lost). The top panel shows a case with average starting parallel velocity, the middle panel shows a case with parallel velocity 1 standard deviation below the average, and the bottom panel shows a case for parallel velocities 1 standard deviation above the average. We see that black lines do not show any reasonable behavior outside the magnetosphere since the T96 magnetic model fails outside the magnetosphere. Consequently, the traces are unreliable but the ions definitely end up in the magnetosheath. Most of the oxygen ions escape into the solar wind beyond the distant X-line. A fraction of the oxygen ions is convected to the plasma sheet, and a small part will escape into dayside magnetosheath. From our results, it takes 120 minutes on average for oxygen ions to reach distant X-line (based on average parallel $\langle v_{par} \rangle$ velocities). That means that if oxygen ions are not convected into the plasmasheet within 120 minutes they will most likely escape beyond the distant X-line.

In figure 9 we show the results of the tracing on the sampling bins (starting positions of the tracing) i.e. the colors indicate where the tracing will end starting from each bin. Colors used are the same as in figure 8. The average cusp ion outflow is $3.9 \times 10^{24}$ s$^{-1}$ and the estimated percentages of oxygen flux which end up in each region are given in table 2.

From our estimation, on average 31 % of the total oxygen flux from the high altitude cusp gets convected to the plasma sheet. The further fate of these ions and transport inside the plasma sheet is beyond the scope of this paper, but it is reasonable to assume that a fraction of the recirculated ions are eventually lost through plasmoid ejections, through the magnetopause and other loss processes.

We also present the resulting oxygen outflow for different storm conditions, using the Dst index as a proxy for storm conditions. For quiet conditions we used positive Dst values, for moderate storm conditions we used Dst values between 0 and $-20$ nT, and for active storm conditions we used Dst values below $-20$ nT. For quiet and active storm conditions for nightside measurement bins ($X_{GSM} \leq -1$ $R_E$) the coverage is rather poor, but this is not a major problem, since the oxygen fluxes are rather low under these conditions, thus not affecting the overall results significantly. The threshold for active storm conditions might seem a bit high, but for a lower threshold we have a much smaller dataset and a lot more data gaps (see appendix figures

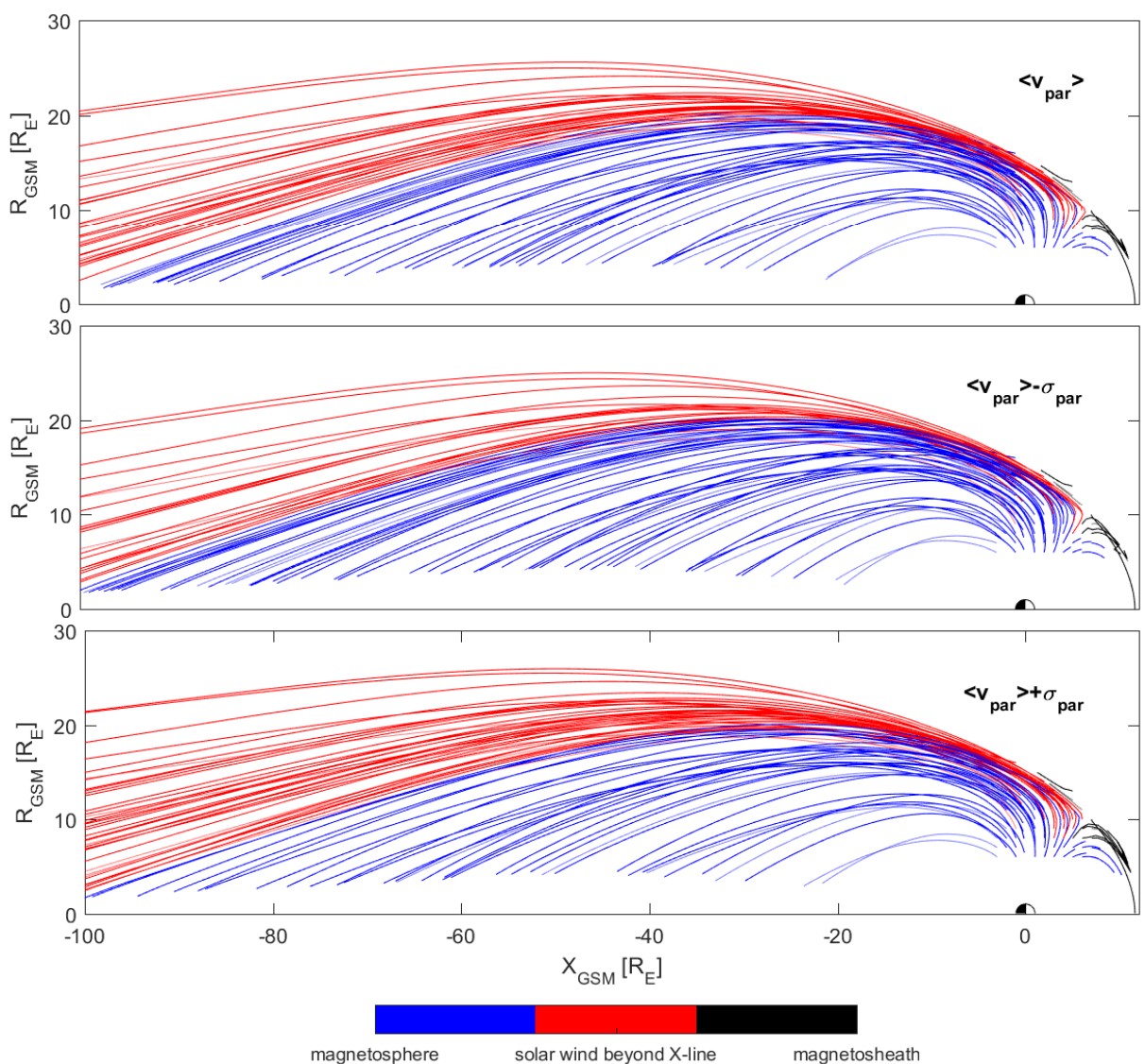

**Figure 8.** Tracing results using initial parallel velocities. Individual lines show the paths of particles from each measurement bin. Colors indicate the fate of oxygen ion: Blue indicate that they will return back to magnetosphere (mostly plasma sheet), red color indicate ions ending up in the solar wind, black indicate ions escaping into the dayside magnetosheath. Different panels represent cases for different starting velocities: The top panel shows results using average velocities, middle panel shows results using lower standard deviation velocities and the bottom panel shows results using upper standard deviation velocities.

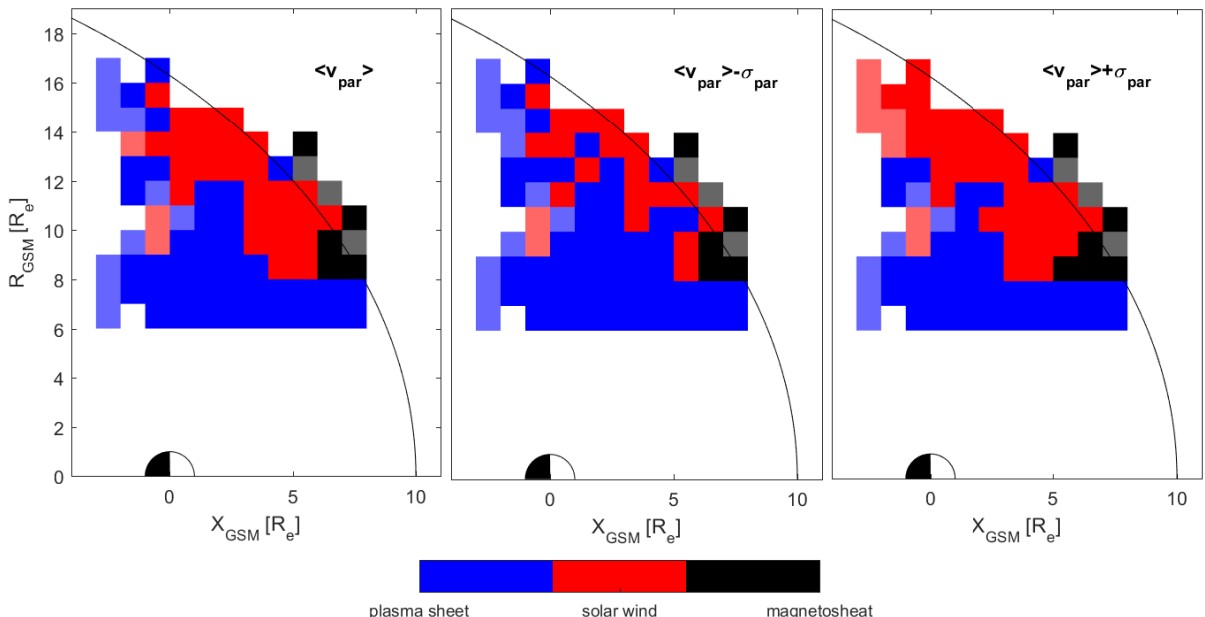

**Figure 9.** The figure depicts the results of tracing for each starting bin. The different panels show various starting parallel velocities. Cases for the starting parallel velocities form left to right are: average parallel velocity, lower standard deviation parallel velocity and upper standard deviation parallel velocity. Transparent colours represent the bins with less than 30 minutes of EDI measurement.

A1-A3). The results of the tracing for different storm conditions are given in Figure 10. As seen from this figure the results are highly dependent on storm conditions. The gaps for active and quiet conditions would probably favour capture, since for moderate storm conditions this regions bins all show capture, but it would only change results by a couple of percent, due to the lower fluxes in those bins (See figure 7). The most interesting case is the tracing during active storm conditions, because most of the outflowing oxygen flux gets convected into the plasma sheet. During strong storms, both parallel velocity and convection velocities increase, but the increase in convection is stronger, causing a larger flux of oxygen ions into plasma sheet. The outflowing $O^+$ ions are deposited closer to Earth, for storm geomagnetic conditions. In figure 11 we show the results of the tracing in starting bins in the same way as in figure 9, but for various geomagnetic conditions. The estimated percentages of fate of oxygen flux for various Dst conditions are given in table 3.

## 5 Discussion

In terms of oxygen outflow escape from the high altitude cusps and plasma mantle regions we find that most of the oxygen escape the magnetosphere, as shown by Slapak et al. (2017). As pointed out by Seki et al. (2002) and Ebihara et al. (2006), oxygen ions with low energies ($< 1\,\mathrm{keV}$) will end up in near tail plasma sheet or in ring current. Our results show that oxygen ions reaching the high altitude cusps will mostly escape the magnetosphere. On average, $50\%$ of the oxygen outflow flux will

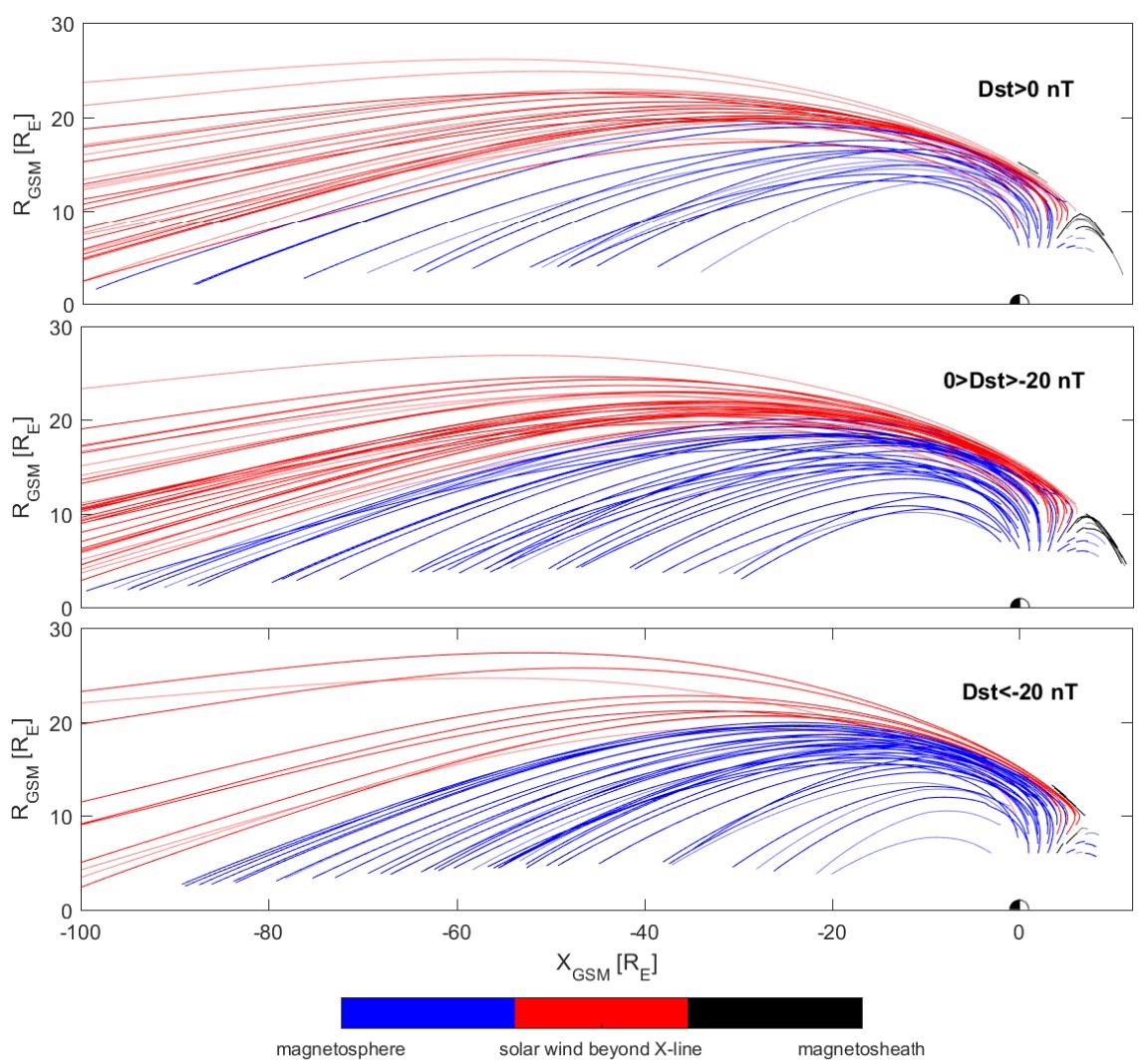

**Figure 10.** The tracing results using parallel initial velocities for different storm conditions. The upper panel shows quiet conditions, middle panel shows moderate storm conditions, and the lower plot shows active storm conditions. The colors are the same as in fig. 8

**Table 3.** Estimated fate of oxygen ions expressed as percentages of outflow flux. $\Phi$ represents the flux, and subscripts $ms$, $sw$ and $ps$ represent magnetosheath, solar wind and plasmasheet, assuming that the plasmasheet is limited by distant X-line at $X_{GSE} = -100\,\mathrm{R_E}$

|            | $Dst > 0\,\mathrm{nT}$ | $0 > Dst > -20\,\mathrm{nT}$ | $Dst < -20\,\mathrm{nT}$ |
|------------|------------------------|------------------------------|--------------------------|
| $\Phi_{ms}$ | 9 %                   | 20 %                         | 12 %                     |
| $\Phi_{sw}$ | 62 %                  | 50 %                         | 15 %                     |
| $\Phi_{ps}$ | 29 %                  | 30 %                         | 73 %                     |

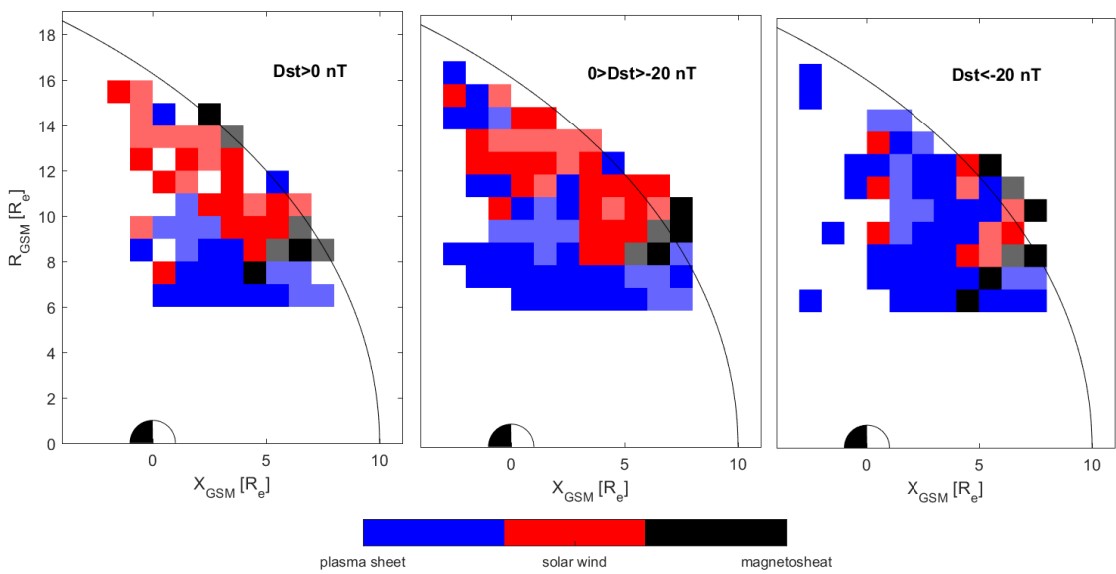

**Figure 11.** Results of tracing for each starting bin. Different panels depict various starting geomagnetic conditions. Cases for the starting parallel velocities form left to right are: quiet condition, moderate condition and active geomagnetic condition. The colors are the same as in figure 8

end up in the solar wind beyond the distant X-line. 19% will escape directly into the dayside magnetosheath. This sums up to a total escape rate of 69 % of high altitude cusp oxygen flux. The rest, 31 % of the high altitude cusp flux is being convected to plasma sheet, mostly in the distant tail ($> 50$ $R_E$), as shown by the figure 8.

Another important issue is the escape-versus-capture ratio for different storm conditions. During quiet magnetospheric conditions, oxygen outflow and energization is relatively low, resulting in lower fluxes of oxygen in the high altitude cusp. However, in such cases, the magnetospheric convection is also low and consequently almost all of the outflowing oxygen escape. It is worth mentioning that in such cases IMF is mostly northward and can lead to lobe reconnection, resulting in sunward flow. This process can decelerate oxygen ions, and lead to their capture. Positive Dst periods are also often characterised by sudden

high $P_{DYN}$ changes (e.g., Boudouridis et al., 2007; Gillies et al., 2012). We did not did not differentiate sudden commencement form quiet conditions, since we used the average values of large dataset. We assume that such changes are increasing our average convection velocities and the oxygen outflow (which are low), and without them the results would not change much all together. During moderate storm conditions, results are similar to average conditions. For active storm conditions, the oxygen ion flux is high, and both the parallel velocity of the oxygen ions and the convection is higher. This leads to increase in both

dayside magnetosheath escape and enhanced convection into the plasma sheet. Oxygen ions are more likely to escape into the dayside magnetosheath due to their high parallel velocities. Oxygen ions that get convected from the cusps into the plasma mantle will eventually be convected into the plasma sheet. There are also other processes which can further energize ions on

their path during strong magnetospheric storms, and thus cause them to escape beyond X-line. For example Lindstedt et al. (2010) reported additional energization of few keV at cusp-lobe boundary during strong geomagnetic storms, caused by increased reconnection leading to strong localised Hall electric field and non adiabatic motion of the ions.

Lennartsson et al. (2004) reported observations of oxygen ions with energies of 3-4 keV in the magnetospheric lobes around $10 \, \mathrm{R_E}$ during geomagnetic storms. In our tracing, ions with such high energies in the tail around $10 \, \mathrm{R_E}$ are traveling close to the magnetopause, and the results of Lennartsson et al. (2004) cannot be verified by our study . During geomagnetic storms, 73 % of the oxygen flux end up in the plasmasheet, but far down in the tail (beyond $50 \, \mathrm{R_E}$). The high energy oxygen ions in the lobes reported by Lennartsson et al. (2004), are more likely the result of magnetospheric energization of existing low energy oxygen ions in the lobes, rather than convection of high energy oxygen ions. The overall dependence of oxygen capture during storm conditions agrees with results from Haaland et al. (2012), in the sense that we observe increased capture during active storm conditions, and more escape during quiet conditions. The main difference is that Haaland et al. (2012) analyzed capture rate of low energy hydrogen ions in the lobes emanating from the polar cap regions, while in this paper we have analyzed the fate energy oxygen ions emanating from the cusp regions.

## 6 Conclusions

In this paper we have used Cluster EDI data in the lobes in combination with the CODIF cusp dataset from Slapak et al. (2017), to obtain parallel and convection velocities for oxygen ions. Furthermore, we used results from Nilsson et al. (2006, 2008) for accelerations in cusps and lobes, as well as results from Newell and Meng (1987) for cusp width, to estimate the fate of oxygen ions originating from the high altitude cusp regions. The findings are summarized as follows:

1. Assuming that the magnetosphere terminates at a distant X-line fixed at $X = -100 \, \mathrm{R_E}$, 69 % of total oxygen outflow from the high altitude cusps escape the magnetosphere on average. 50 % escape tailward beyond distant the X-line and 19% escape to the dayside magnetosheath.

2. The oxygen capture-versus-escape ratio is highly dependent on geomagnetic conditions. Oxygen ions originating in the cusp are more likely to be captured during active conditions since the majority of oxygen outflow is convected to plasma sheet, although rather far downtail.

3. The average time for oxygen ions to reach distant X-line ($-100 \, \mathrm{R_E}$) is 120 minutes.

**Appendix A: Data distribution for various storm conditions**

Figures A1, A2, A3 show the datasets we used for different storm conditions. The left panel in each figure shows the CODIF dataset and the right panel shows the EDI dataset. Red vectors represent average vector in each sampling bin, scaled with the vector shown in lower right corner in each panel. Colour of the bin represent the number of one minute data in that bin. Figure

A1 shows the data distribution for quiet geomagnetic conditions, figure A2 shows data distribution for moderate geomagnetic conditions, and figure A3 for active geomagnetic conditions.

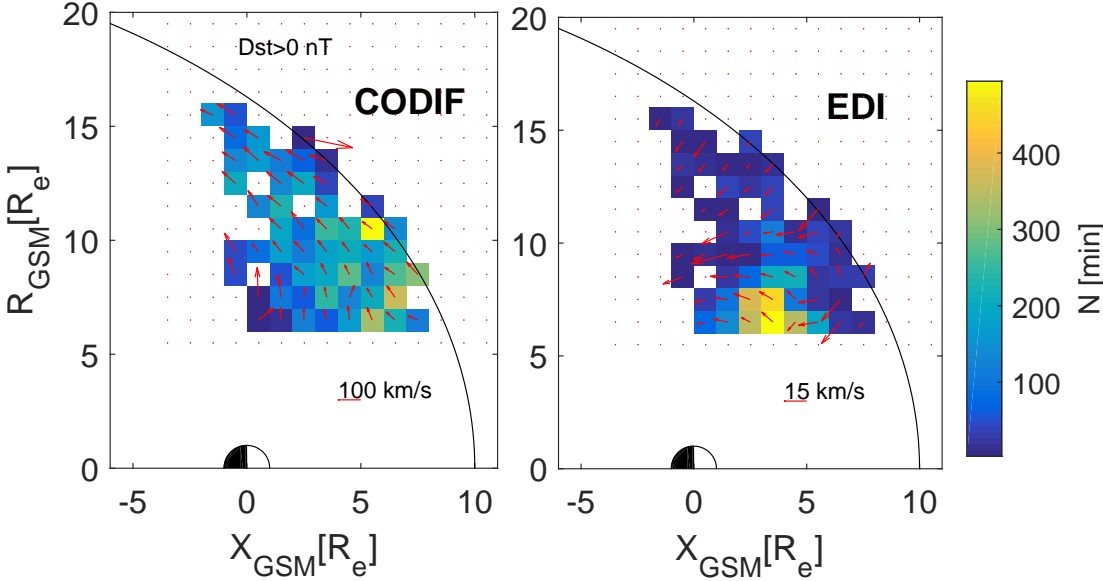

**Figure A1.** Distribution of average parallel and convection velocities in the cusps and plasma mantle regions for quiet geomagnetic conditions. Lengths of vectors represent convection velocity in each bin, calculated as the average magnitude of vectors. Colours indicate the number of one minute measurements in each bin. Left panel depicts parallel velocities obtained using CODIF data; right panel depicts convection velocities obtained using EDI data. Vectors are scaled as given in the lower right corner of each panel.

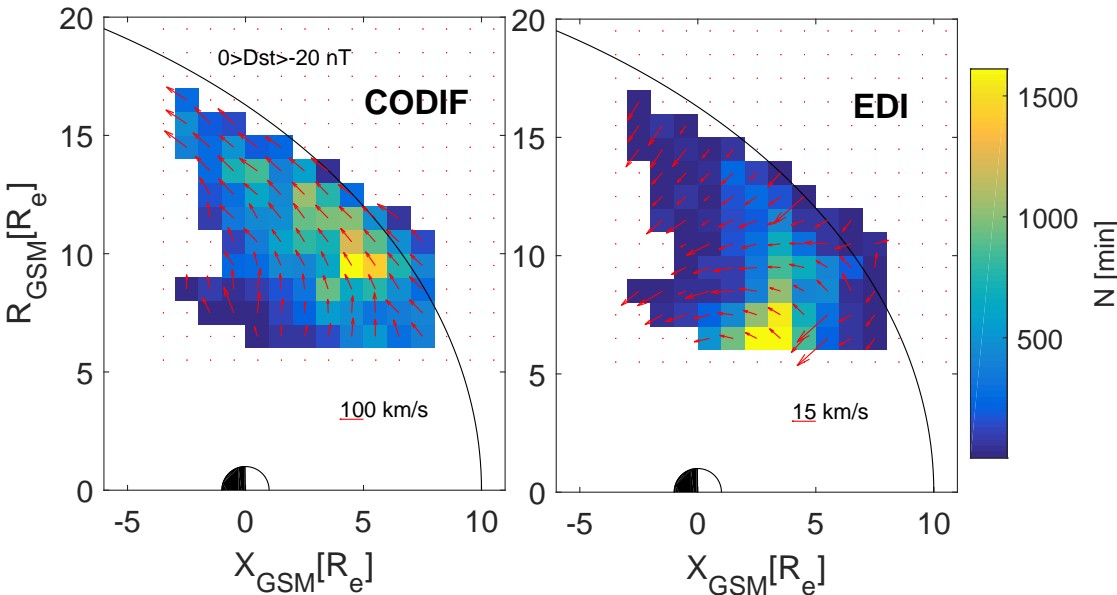

**Figure A2.** Distribution of average parallel and convection velocities in the cusps and plasma mantle regions for moderate geomagnetic conditions. Labels are the same as in figure A1.

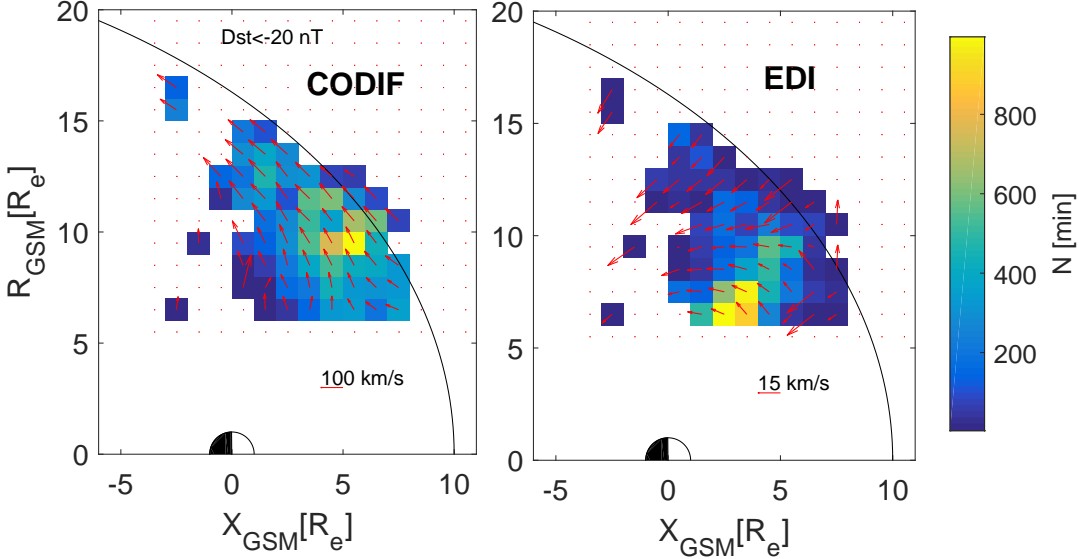

**Figure A3.** Distribution of average parallel and convection velocities in the cusps and plasma mantle regions for active geomagnetic conditions. Labels are the same as in figure A1.

*Author contributions.* P.K. and S.H. conceived the presented idea. P.K. analysed EDI data and performed the oxygen ion tracing. R.S and A.S analysed and prepared the CODIF data. S.H. supervised the project. P.K. improved the method in discussions with S.H. and L.M. All authors contributed to discussion and P.K wrote the paper with input from all authors.

*Competing interests.* The authors declare that they have no conflict of interest.

5 *Acknowledgements.* P. K. acknowledges support from University of Zagreb, Croatia, Erasmus+ program. P. K. thanks the Max Planck Institute for Solar System Research, Göttingen, Germany for hosting him during the duration of this work. S. H. were supported by the Norwegian Research Council (NFR) under Grant 223252 and the German Aerospace Center (DLR) under contract 50 OC 0302. L. M. acknowledges the support by the Deutsches Zentrum für Luft-und Raumfahrt, grant DLR 50QM170. R. S. acknowledges the finantial support from EISCAT Scientific Association, Kiruna, Sweden. A. S. acknowledges the financial support from the Swedish Institute of Space Physics,
10 The Graduate School of Space Technology in Luleå, Sweden. The Cluster EDI and CODIF data is provided by ESA's Cluster Science Archive, and can be accessed at http://csa.esac.esa.int/csa-web/. We acknowledge use od NASA/GSFC's Space Physics Data Facility's OMNIWeb service, and OMNI data, which can be accessed at http://ominweb.gsfc.nasa.gov. P.K. would like to thank the whole MPS group for all their support and for making this paper possible.

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
