# Peer review of "Estimating the fate of oxygen ion outflow from the high altitude cusp"

_Annales Geophysicae, 2019_

## Short Comment (SC1) · 3 Sep 2019

Patrik Krcelic, Stein Haaland, Lukas Maes, Rikard Slapak, and Audrey Schillings Ann. Geophys. Discuss., https://doi.org/10.5194/angeo-2019-125, 2019 "Estimating the fate of oxygen ion outflow from the high altitude cusp"
=============================================================
I think, this is an interesting statistical study, that makes use of Cluster EDI and CODIF data to investigate the fate of escaping oxygen ions under various conditions. It should be published in any case.

I have, however, some remarks and found a few misprints (see below).

In section 2.2, related to Fig. 1, I'm missing a statement about the IMF (orientation,

[Figure]

strength) assumed. The IMF orientation is also important for the precise position of the cusp; the symmetric position about noon suggests, that you assumed By=0. The exact cusp positioning would be better described (I suppose) by the newer Tsyganenko models. But I agree, that for a first, simpler estimation the older T96 (with no IMF-dependence?) is sufficient.

Fig. 2: "...projected into the Northern Hemisphere..." - how this projection is done? The majority of data ($\sim$2/3) are projected ones. Maybe, one should rather generally speak about projected data, because they are shown in a x-R plane, where "R" is by definition always positive, independently of Northern or Southern provenience.

Fig. 6, right panel: the convection should actually be perpendicular to B; this is quite strongly disturbed close to the Shue-et-al model magnetopause, but also somehow in other regions. This is clearly due to the averaging within the pixels and an indication for the variability of the convection data in this region. The later estimates probably take only ratios between parallel and perpendicular velocity components, not the (slightly "chaotic" or "random") data?

Page 9, paragraph below Fig. 5, V_i perp: "...the scaling of cusp convection ...illus-trated in Fig. 5." Why only "illustrated"? One can quantify it by the (inverse) ratio of |B| strength between ionosphere and magnetosphere, unless I'm mistaken. Why not specify this here?

Minors:

Caption of Fig. 1, 3rd sentence: some small words are missing - I'd write: "The right panel depicts the schematic (symmetric) area that the cusp and plasma mantle occupy in the polar cap."

"magnetosheath" with end-"h" in captions to Fig. 3, in the labels of Figs. 9 and 11, and on line 5 on page 17.

Page 6, line 6: "...has a wider temperature range."

Page 6, line 27: "In the present paper..."

Page 7, line 17: "considered" (with "ed")

Page 9, line 9: I would write in plural: "...length of the arrows indicates the magnitude of the vectors.." And: the scale in Fig. 6 is NOT in the upper right corner, but below the binned area.

Page 11, line 10: instead of "an" -> "the" ...measurements...

Page 15, line 1: include "the" in "... of the fate..."

Page 15, line 5: order of words: "...from the high latitude..."

Page 16, line 8: include "ions" in "...oxygen ion escape."

Page 17, line 15: "...ends up..." Page 17, line 19: "...analyzed the capture..." line 21: "...fate of energy oxygen ions..." (or energetic?)

Conclusions, first item: order of words: "...the distant X-line..." 72,1 Bo

---

## Referee Comment (RC1) · Anonymous Referee #1 · 18 Sep 2019

**General comments**

This paper studies oxygen ion escape from the polar cusps and determines which region ions ultimately reach (dayside magnetosheath, plasma sheet, or solar wind through the distant X-line in the magnetotail), and in which proportions. The authors use measurements from the EDI and CODIF instruments aboard Cluster to obtain statistics of cusp and plasma mantle convection speed and oxygen ion fluxes and parallel speeds, respectively. They trace the trajectories of oxygen ions from their initial location in the cusp in the geomagnetic field using the Tsyganenko and Stern T96 model, with a simple parametrisation for particle acceleration along geomagnetic field lines depending on the region in the cusp/lobe. The conclusions are that, over-all, 69% of the oxygen escaping from the high-latitude cusps reaches the solar wind

(19% through the dayside magnetosheath, 50% through the magnetotail distant X-line), whereas 31% reaches the plasma sheet. When separating driving conditions as a function of geomagnetic activity given by the Dst index, the authors find that enhanced activity (Dst < −20 nT) increases the amount of oxygen ions returning to closed field lines in the plasma sheet.

The paper is pleasant to read, as it is well-organised and has clear figures. The authors carefully justify their choices such as the magnetic field model version or the cusp latitudinal extent. My comments below are therefore mostly minor ones, except perhaps regarding the choice of Dst ranges for binning according to geomagnetic conditions.

**Specific comments (major)**

The only major comment I have is related to the choice of Dst values used in the data separation according to geomagnetic activity. Indeed, positive values of Dst can be associated with so-called sudden commencement, which is generally triggered by solar wind pressure pulses (see for instance https://doi.org/10.1029/2006JA012141; https://doi.org/10.1029/2011JA017255). This seems consistent with the fact that, in Table 1, the solar wind dynamic pressure has its highest average value for the Dst > 0 condition. Therefore, it seems a bit strange to me to call this category "quiet conditions" since they may include sudden commencement events.

On the other hand, the threshold of Dst < −20 nT to define "active" conditions would need to be justified. Indeed, Dst = −20 nT is considered by some authors to belong to the category of "quiet conditions" (see for instance https://doi.org/10.1029/2008JA013095 paragraph [30] in sect. 5.3.1).

I would therefore recommend to either modify the Dst thresholds when studying the effects of geomagnetic activity on the fate of oxygen ions, or to add a justification for choosing these values as well as a discussion on potential effects of sudden commencements in the Dst > 0 situation.

**Specific comments (minor)**

- p. 2, l. 21: It could be helpful for the reader to give a range of values for "high altitudes", since it is a recurring concept in this paper.

- Figure 2: Perhaps it would be nice to add the lines showing the boundaries of cusp and plasma mantle from Fig. 1 on both panels of Fig. 2, for the reader to be able to relate it to the discussion on p. 5, l. 3.

- p. 8: I would suggest that, whenever writing acceleration units, a space be added ($m\ s^{-2}$) to avoid any possible confusion with "per squared millisecond".

- Figure 4: I would recommend slightly increasing the line thickness as well as the font size to improve the legibility of this figure.

- p. 12, l. 8–9: Strictly speaking, the lower (upper) quartile is not defined as one standard deviation below (above) the average. I would therefore recommend to rephrase the corresponding sentence.

- Figure 9: I am intrigued by the isolated black pixel in the right panel (near X = 1.5 $R_E$, R = 8.5 $R_E$). It is mostly curiosity, but would it be possible to briefly explain the reason why the oxygen from this bin reaches the dayside magnetosheath rather than the plasma sheet?

- p. 14: Would it be possible to add, perhaps as supplementary material, a figure showing the coverage of EDI and CODIF data for each of the three geomagnetic activity conditions, in a similar way as shown in Fig. 2? This could prove useful when discussing the results shown in Fig. 11. While the rather poor coverage for "quiet" and "active" conditions is unlikely to affect the overall results (as stated on p. 14, l.3), it would be valuable to discuss to what extent conclusions on the oxygen fate under those two types of conditions might be affected by the lack of data.

- p. 18, l. 1: The third point given in conclusions has not been mentioned above. I think it would be better to introduce the corresponding result and explain how it was obtained in one of the previous sections (Results or Discussion).

**Copyediting and typesetting**

- p. 5, l. 11: ware → were

- p. 7, l. 17: consider → considered

- p. 11, l. 10: an measurements → a measurement / measurements

---

## Referee Comment (RC2) · Anonymous Referee #2 · 2 Oct 2019

SUMMARY This work uses several years of in-situ observations made by Cluster to estimate the relative amount of O+ cusp outflow that is lost versus the amount that is re-circulated in the Earth's magnetosphere. They use a combination of two instruments to infer the average parallel and convection velocity of O+ in the cusps. Then, they compute whether the averaged outflows should end up in the dayside magnetosheath, distant tail beyond 100 R_e, or closer tail. They find that 52 - 82 % (table 3) of the O+ outflow is lost, and the rest is re-circulated to the magnetosphere. They also discuss their results as a function of Dst index.

Overall evaluation

Main comments

[Figure]

It is not clear to me why you try to restrict to cusps and plasma mantle outflows. Is your method not valid outside of these regions? Why?

Your results are provided as a function of position in GSM. They way you ensure they correspond to cusp / plasma mantle is not fully clear to me (see below).

CODIF obtains a full 3D velocity vector. Why do you choose to use v_par from CODIF and ExB~v_perp from EDI? You could use v_perp from CODIF instead, right? I agree this assumes that O+ is frozen-in, which may not be always the case. At the very least, you should compare v_perp from EDI with v_perp from CODIF when both measurements are available, and maybe also with v_perp from HIA. I would be curious to see if your Figure 6(right) is very different when computed using v_perp from CODIF or HIA.

Another main concern to me is if the dataset you use corresponds truly to cusps observations. For EDI you use TS96 to decide if you are in the cusps or not only, right? You should check other parameters as well when available, as for instance plasma beta. For CODIF dataset you do a much more accurate filtering of your dataset.

English grammar needs to be revised.

Overall, I find interest in this study, but further clarifications are required in the text, figures and tables. The Introduction and referencing may be missing relevant recent studies.

Detailed comments

Introduction. Global models, eg Glocer et al. 2009 (Modeling ionospheric outflows and their impact on the magnetosphere, initial results) should be discussed somewhere in the manuscript.

Other works that potentially should be cited, discussed and compared to this study:

Slapak and Nilsson 2018 'The Oxygen Ion Circulation in The Outer TerrestrialMagnetosphere and Its Dependenceon Geomagnetic Activity'

Liao et al. 2010 'Statistical study of O+ transport from the cusp to the lobes with Cluster CODIF data'.

P2.5 remove comma

P2.8 'has been analyzed' in detail?

P2.12 For given solar wind condtions...

P2.13 The convection velocity scales with magnetic field –> with the inverse of magnetic field magnitude?

P3. 11-13 In which parameter space? Dst? GSM coordinates?

P4.3 Please include the reference to the newer model that you decide not to use, for completeness.

P5.1 '448 hours are from the cusps'. D you infer cusp/no cusp of each 1 min EDI measurement using TS96 with its corresponding Kp index? Could you be a bit more precise on how do you get this number?

P5.4 'good quality EDI'. Can you specify you criteria for 'good quality'?

P5.8 Do you impose R> 6 R_E as for CODIF? Please specify.

P5.9 Please include the parameters you used for computing Shue98 (Pd and Bz).

P5.11 'ware'

P5.14 'Jan-June' Is this because during July-Dec Cluster does not cross the region of interest for this study? Would be equivalent to say you used all available data in 2001-2005? Please clarify if there is another reason to use Jan-June only.

P5.17 How do you get beta? Do you use CODIF or HIA for the ion pressure? Do you account for the contributions of all species or only H+?

P6.5 O+ densities in both...

P6.1-9 The description of the method to choose CODIF data is a bit confusing. You do not mention the word 'cusps', but this is the O+ population you are interested in, right? Plasma mantle <-> cusp, here? Which energy range of CODIF do you use to compute v_par? Is cluster in the cusps/plasma mantle according to TS96 for all the measurements you select from CODIF?

P6.19 Could you comment on the drawbacks of this criterium (100 R_E)? The X line position is not well defined, and can be significantly lower during disturbed conditions.

P6.18-30 I do not understand how do you 'trace' your outflows. Could you explain a bit more what you (Haaland, Li) do for propagating the outflows to the tail?

P7.8 will retain

P7.17 considered

P8.7 'in the cusp regions'. Based on TS96?

P10.12. this is a very crude simplification, although I understand it is difficult to do better given the current knowledge of the distant tail. The shortcomings of this approach need to be discussed, though.

P11.3. To me, a very interesting result would be what is the average O+ flux in the cusps. Why do you not give this number and prefer to give relative amounts only? Slapak et al. 2017 does provide this number, right? Please include it also in this manuscript. It would be interesting also to see how it compares to other independent estimations of the O+ outflow in the cusps.

P11.10 an measurements

Table 1. You average over many years of data. I recommend including std deviation to these quantities, which I suspect may be large.

P12.8-9 'Quartile' is not appropriate here.

P15.5 the high-altitude

P15.6 ..., as shown by...

Table 3. The consideration XGSE = -100 R_E may not be accurate for high-activity (Dst < -20 nT) periods. Include Dst units.

P17.5 magnetosheath
* * *

---

## Author Comment (AC1) · 8 Nov 2019

[12pt,a4paper]article xcolor

**Answers to referee**

We thank the reviewer for careful reading of the manuscript, and for providing valuable suggestions for improvement. Straightforward changes such as grammar, are changed in manuscript, and are not additionally commented. In following sections we firstly repeat the comments from reviewer, and than in section "Authors response" provide our response to the comments.

**1  Comments and Authors response**

**1.1  Major**

The only major comment I have is related to the choice of Dst values used in the data separation according to geomagnetic activity. Indeed, positive values of Dst can be associated with so-called sudden commencement, which is generally triggered by solar wind pressure pulses (see for instance $https : //doi.org/10.1029/2006JA012141$; $https : //doi.org/10.1029/2011JA017255$). This seems consistent with the fact that, in Table 1, the solar wind dynamic pressure has its highest average value for the $Dst > 0$ condition. Therefore, it seems a bit strange to me to call this category "quiet conditions" since they may include sudden commencement events. On the other hand, the threshold of $Dst < --20\,nT$ to define "active" conditions would need to be justified. Indeed,

$Dst = --20 \ nT$ is considered bysome authors to belong to the category of "quiet conditions" (see for instance $https: //doi.org/10.1029/2008JA013095$ paragraph [30] in sect. $5.3.1$).I would therefore recommend to either modify the Dst thresholds when studying the effects of geomagnetic activity on the fate of oxygen ions, or to add a justification for choosing these values as well as a discussion on potential effects of sudden commencements in the $Dst > 0$ situation.

We did not take the sudden commencement into account in our Dst separation, but rather looked for the average behaviour for various Dst range. For instance the average convection velocities and oxygen fluxes are lower for $Dst > 0 \ nT$ than for negative Dst values and thus we have named it the "quiet" condition. The sudden commencement toes indicate the beginning of a storm and should not be classified as "quiet", but the convection (and thus most of the effects of the storm on ion outflow, like centrifugal forces etc) has not yet been set up, so this can be justified. If needed the classification name "quiet conditions" can be changed in manuscript. The short discussion on the sudden commencement with your recommended references and it effects on the results are now added into the manuscript under the section "Discussion". For "active" conditions the main reason for choosing the $Dst = -20 \ nT$ threshold is in the statistics. If we chose the lower value for the Dst threshold we have considerably less data to make our analysis, and a lot more gaps occur. The average Dst value in our "active" dataset is $-41$, because of the extreme values going up to $-200 \ nT$. But again the average convection velocities and the oxygen fluxes are higher than for our "average" conditions and therefore we have named it "moderate" conditions. The short explanation for $Dst = -20 \ nT$ threshold is now added into the manuscript in section "Results" where the storm separation of the data is first introduced.

**1.2 Minor**

p. 2, l. 21: It could be helpful for the reader to give a range of values for "high altitudes", since it is a recurring concept in this paper.

High altitude cusps do not have a defined boundary, but in this paper we used values with $R > 6\ R_E$, and are added into the text.

Figure 2: Perhaps it would be nice to add the lines showing the boundaries ofcusp and plasma mantle from Fig. 1 on both panels of Fig. 2, for the reader to be able to relate it to the discussion on p. 5, l. 3.

In this figure are the data taken from cusps and plasma mantle over a long period of time ($4$ years for CODIF and $14$ years for EDI data). The boundaries are constantly changing due to a dynamic nature of the cusps, and adding the some average boundaries might be a confusing to some readers, since much of the data would be outside of this average boundaries.

p. 8: I would suggest that, whenever writing acceleration units, a space be added(m s$-2$) to avoid any possible confusion with "per squared millisecond".

I agree and have changed it.

Figure 4: I would recommend slightly increasing the line thickness as well as the font size to improve the legibility of this figure.

Line thickness and font size are now increased.

p. 12, l. 8–9: Strictly speaking, the lower (upper) quartile is not defined as one standard deviation below (above) the average. I would therefore recommend to rephrase the corresponding sentence.

The sentence is changed and we do not mention the quartiles at all, only standard deviations.

Figure 9: I am intrigued by the isolated black pixel in the right panel (nearX = 1.5RE, R = 8.5RE). It is mostly curiosity, but would it be possible to briefly explain the reason why the oxygen from this bin reaches the dayside magnetosheath rather than the plasma sheet?

The isolated pixel in the figure 9, is an error in our code which occurred in the ploting part of the code. Thank you for pointing it out, the code has been fixed. The results and conclusions are the same.

p. 14: Would it be possible to add, perhaps as supplementary material, a figure showing the coverage of EDI and CODIF data for each of the three geomagnetic activity conditions, in a similar way as shown in Fig. 2? This could prove useful when discussing the results shown in Fig. 11. While the rather poor coverage for "quiet" and "active" conditions is unlikely to affect the overall results (as stated on p. 14, l.3), it would be valuable to discuss to what extent conclusions on the oxygen fate under those two types of conditions might be affected by the lack of data.

The requested figures are now added to the appendix. The gaps in the coverage are in the region that is almost entirely convected in "neutral" condition panel. We assume that if we do not have this gaps the results might favour the "capture" by few percent. This comment is now added into section "Results" of the changed manuscript.

p. 18, l. 1: The third point given in conclusions has not been mentioned above.I think it would be better to introduce the corresponding result and explain how itwas obtained in one of the previous sections (Results or Discussion).

The third point in conclusion is now added into section "Results" in the changed manuscript.

**Supplement:**

[revised manuscript text omitted]

---

## Author Comment (AC2) · 8 Nov 2019

[12pt,a4paper]article xcolor

**Answers to the referee**

We thank the reviewer for careful reading of the manuscript, and for providing valuable suggestions for improvement. Straightforward changes such as grammar, are changed in manuscript, and are not additionally commented. In following section we firstly repeat the comments from reviewer and than provide our response to the comments.

**1   Comments and Author response**

**1.1   Main comments**

It is not clear to me why you try to restrict to cusps and plasma mantle outflows. Is your method not valid outside of these regions? Why?

Method is valid in the polar cups as well, but in the polar caps the oxygen ion energies (and therefore parallel velocities) are much smaller and the ions get captured in the near Earth plasmasheet and ring current. The main concern of this paper is, what happens with oxygen ions from cusps and plasma mantle as it is thought that they all escape the magnetosphere.

CODIF obtains a full 3D velocity vector. Why do you choose to use $v_{par}$ from CODIF and $E \times B v_{perp}$ from EDI? You could use $v_{perp}$ from CODIF instead, right? I agree this assumes that O+ is frozen-in, which may not be always the case. At the very least,you should compare $v_{perp}$ from EDI with $v_{perp}$ from CODIF when both measurements are available, and maybe also with $v_{perp}$ from HIA. I would be curious to see if your Figure 6 (right) is very different when computed using $v_{perp}$ from CODIF or HIA.

There is a big difference between the EDI and CODIF perpendicular velocity data. The CODIF perpendicular velocities have similar values to CODIF parallel velocities. This velocities go up to $120\ km/s$, and are definitely not from the convection. EDI data give values of around $15\ km/s$ which is what we expect convection to be. At this point we do not know how to explain the CODIF perpendicular velocity measured in the cusps.

Another main concern to me is if the dataset you use corresponds truly to cusps observations. For EDI you use TS96 to decide if you are in the cusps or not only, right? You should check other parameters as well when available, as for instance plasma beta.For CODIF dataset you do a much more accurate filtering of your dataset.

The plasma beta number is not always available when we have EDI data. We have decided to analyse each dataset separately and than combine the average values to get our estimate.

**1.2   Detailed comments**

Introduction. Global models, eg Glocer et al. 2009 (Modeling ionospheric outflows and their impact on the magnetosphere, initial results) should be discussed somewhere in

the manuscript. Other works that potentially should be cited, discussed and compared to this study:

Slapak and Nilsson 2018 'The Oxygen Ion Circulation in The Outer Terrestrial Magnetosphere and Its Dependenceon Geomagnetic Activity'

Liao et al. 2010 'Statistical study of O+ transport from the cusp to the lobes with Cluster CODIF data'

The mentioned papers are now added to introduction.

P3. 11-13 In which parameter space? Dst? GSM coordinates?

Yes we used GSM coordinates and Dst values to combine the data. We have decided to remove the phrasing "parameter space" in manuscript to avoid confusion.

P4.3 Please include the reference to the newer model that you decide not to use, for completeness.

The references to the newer models are now included.

P5.1 '448 hours are from the cusps'. Do you infer cusp/no cusp of each 1 min EDI measurement using TS96 with its corresponding Kp index? Could you be a bit more precise on how do you get this number?

Yes, we have labeled all EDI minute measurements as "cusp/no cusp" using T96 model, and got the total number of the one minute measurements inside cusps. The better explanation is given in the new version of the manuscript.

P5.4 'good quality EDI'. Can you specify you criteria for 'good quality'?

"good quality EDI" is an label given by "Cluster Science Archive (CSA)", and there are a series of the criteria explained in the doi$= 10.1007/978 - 90 - 481 - 3499 - 1_5$;. The criteria are mostly statistical ($\chi^2$ analysis is the most important one), and most of the scientific work is done using this data without getting too much into other two labels "caution" and "bad" data. A short explanation is included in the manuscript.

P5.8 Do you impose $R > 6\ R_E$ as for CODIF? Please specify.

Yes, we impose the $R > 6\ R_E$ as for CODIF and it is now specified in the new version of the manuscript.

P5.9 Please include the parameters you used for computing Shue98 (Pd and Bz).

For Shue98 we used the parameters $B_z = -1\ nT$, and $p_{DYN} = 2\ nPa$, and are now included into the manuscript.

P5.14 'Jan-June' Is this because during July-Dec Cluster does not cross the region of interest for this study? Would be equivalent to say you used all available data in 2001-2005? Please clarify if there is another reason to use Jan-June only.

[Figure]

Yes, during the months Jun-July are the only periods when Cluster crosses the areas of interest. We have changed it to "all available data in years $2001 - 2005$ as you suggested.

P5.17 How do you get beta? Do you use CODIF or HIA for the ion pressure? Do you account for the contributions of all species or only H+?

Plasma beta number is calculated from both $H^+$ and $O^+$ populations, and is it included into the new version of the manuscript.

P6.1-9 The description of the method to choose CODIF data is a bit confusing. You do not mention the word 'cusps', but this is the O+ population you are interested in,right? Plasma mantle $< - >$ cusp, here? Which energy range of CODIF do you use to compute $v_par$? Is cluster in the cusps/plasma mantle according to TS96 for all the measurements you select from CODIF?

For the analysis the full coverage of the CODIF instrument was used, but oxygen ion measurements are in the range $100\ eV$-$4\ keV$. We did not check the data using TS96 model as we did for EDI dataset.

P6.19 Could you comment on the drawbacks of this criterium ($100\ R_E$)? The X line position is not well defined, and can be significantly lower during disturbed conditions.

We have added the comments on the drawback of the position of the distant X-line criterium in the new version of the manuscript.

P6.18-30 I do not understand how do you 'trace' your outflows. Could you explain a bit more what you (Haaland, Li) do for propagating the outflows to the tail?

The method we use is based on the tracing of the ions along the field line using the TS96 model, and moving the field lines with each time step order to simulate the convection. We used the CODIF data to move the ions along the field line in each time step and EDI data to move the field line accordingly. The result is a total path of the ions (along the moving field line).

P8.7 'in the cusp regions'. Based on TS96?

Yes we have here based the cusp regions on the T96 model. This specification is added to the new version of the manuscript.

P10.12. this is a very crude simplification, although I understand it is difficult to do better given the current knowledge of the distant tail. The shortcomings of this approach need to be discussed, though.

The shortcomings of the used regions are now commented in the new version of the manuscript.

P11.3. To me, a very interesting result would be what is the average O+ flux in the cusps. Why do you not give this number and prefer to give relative amounts only? Slapak et al. 2017 does provide this number, right? Please include it also in this manuscript. It would be interesting also to see how it compares to other independent estimations of the O+ outflow in the cusps.

The average values of the cusp oxygen outflow is $\approx 1.05 \times 10^{10}\ m^2\ s^{-1}$, and is now added into the manuscript.

Table 1. You average over many years of data. I recommend including std deviation to these quantities, which I suspect may be large.

The purpose of the "Table 1" is to give the values we have used in our model. Adding the standard deviations into this table might be confusing to some readers.

P12.8-9 "Quartile" is not appropriate here.

Word "quartile" is now removed and the sentence is rephrased.

Table 3. The consideration $XGSE = -100\ R_E$may not be accurate for high-activity($Dst < -20\ nT$) periods. Include Dst units.

Dst units in "Table 3" are now added. The accuracy of specific results are commented in section "Discussion", as the values seems to be to high and is probably not accurate.

**Supplement:**

[revised manuscript text omitted]

---

## Author Response (AR2)

**Answers to referee**

We thank the reviewers for careful reading of the manuscript, and for providing valuable suggestions for improvement. Straightforward changes such as grammar, are changed in manuscript, and are not additionally commented. In following sections we firstly repeat the comments from reviewers, and than in section "Authors response" provide our response to the comments.

**1 Comments and Authors response - Referee 1**

I am overall satisfied with the responses provided by the Authors as well as the corrections which have been implemented to the manuscript. Below are a few additional comments and suggestions after reading the revised version. My recommendation is to accept this manuscript for publication provided those mostly minor comments have been addressed adequately.

p.2 l.21: "> 6 RE <"; I guess the second "<" is a typo.

It was a typo and now is changed.

p.3 l.25: Please define EFW.

EFW is now defined.

p.6 l.25: Could you please specify which time resolution was used for the OMNI data? Given that you use Dst(as opposed to SYM-H), I presume it is the hourly data; is that so?

For $IMF$ and $p_{DYN}$ we used one minute values and for Dst we have implemented a simple linear interpolation in order to get the one minute values. This explanation is now added into the new version of the manuscript.

p.13, l.10: Does the 120 min result come from the analysis using the average parallel velocity $< v_par >$? Or does it combine results from the three analyses $(< v_par >, < v_par > -\sigma_{par}, < v_par > +\sigma_{par})$? I suggest to specify it in the text.

It is a result from $< v_par >$ analysis and is now specified in manuscript.

p. 13 l. 15: There is a typo in the power of 10 (in the ion outflow value).

It is now corrected.

Fig. 8 caption: I think you have forgotten to remove the mention to upper/lower quartiles; please update the caption accordingly.

You are correct and it is now changed.

Fig. 9: Sorry for only noticing it now, but the $< v_par > +\sigma_{par}$ panel seems to have the data shifted one bin "downwards" compared to other two panels. Is this a bug in the code for plotting the figure? Please check, and fix if necessary.

Again, you are correct. It seems that all figures with 3 panels have "downward" shift in third panel. It has now been corrected.

Fig. 9 caption: Same as for Fig. 8, the mention to upper/lower quartiles needs to be removed.

It is now corrected.

p.13 l.24: I suggest to refer to the new figures in the annex (A1, A2, B1).

We agree and the reference to mentioned is now added.

p.15 l.3: How did you estimate that the changes would be "a couple percent" in the absence of data gaps?

We looked at the oxygen flux (figure 7) and noticed that bins with data gaps have an order lower fluxes for order of magnitude compared to the rest of the bins. Therefore, we have concluded that gaps in such bins would not change results much. This explanation is now added into the manuscript.

Fig. 10: I suggest to add the unit (nT) to the Dst values in the top-right-hand corner of each panel.

Dst units are now added into all figures.

Fig. 10 caption: Instead of "labels", shouldn't it be "colors" (which are the same as in Fig. 8)? To me, "labels" would rather refer to the top-right-hand-corner texts about Dst values, which are therefore different from those in Fig. 8. Please also add the unit for Dst values (nT) in the top-right-hand corner of each panel.

You are correct, we have changed the "labels" t "colours" to be consistent with previous descriptions.

Fig. 11: Similarly to Fig. 9, the third panel (Dst ¡ -20 nT) seems to have all its data shifted downwards by one bin; please check and correct if needed.
This is now corrected.

Fig. 11 (also): I am again a bit intrigued by the isolated black pixels. In particular, I notice that in the middle panel (0 ¿ Dst ¿ -20 nT) the isolated black pixel near X = 1 Re, Y = 8 Re is at the same location as the one I was asking about during the first round of comments (in the previous version of Fig. 9, third panel). Since in that first case you found that it was related to a bug in the plotting code, I would recommend that you double check whether the same bug is present in the plotting script of Fig. 11. Regarding the isolated black pixels in the first panel of Fig. 11 (Dst ¿ 0 nT), I suspect that they might be related to the very low number of samples, especially in EDI data (see Fig. A1). In fact, I am a bit worried that quiet conditions have many bins with extremely few (based on Fig. A1, it seems that it could be even less than 10) minutes of observations of convection velocities. How reliable is the analysis when so few observations are available? In that sense, I have the feeling that the discussion on p. 15 l.1-3 needs to be expanded, as presently it only addresses the effect of data gaps but does not mention the cases where the statistics has been obtained with very few measurements. One suggestion I may give is to select a reasonable threshold value (would 30 or 50 min be suitable?) below which the data could still be plotted (provided there are at least 3 minutes of observations, as stated on p.5 l.1) but highlighted as less certain in Fig. 11  for instance, by adding some transparency to the affected pixels. This would not change the conclusions, but could provide some measure of the uncertainty, which I believe would be useful when interpreting the results.

Yes, the isolated black pixel is the result of the bug in the code and is now fixed. We agree with your statement that there seems to be a number of pixels with less than 30-50 min measurements and have decided to implement your suggestion with adding transparency to the figures. We used transparent colors for pixels with less than 30 minute measurements. Thank you for the idea.

Fig. 11 caption: Same as for Fig. 10 caption regarding the word "labels", that I would suggest replacing with "colors".

It is now changed.

p.17 l.15: I am not sure that I understand what is implied by "We did not take those into account" (regarding sudden storm commencement conditions). Do you mean that you removed them from the Dst¿0 dataset, or on the contrary that you did not differentiate them from quiet conditions? Please clarify this statement.

We did not differentiate sudden commencement form quiet conditions. But the low oxygen parallel velocities and low convection velocities for our quiet conditions suggest that sudden commencement do not effect our measurements. The sentence is now altered for more clarification.

Appendix figures: Is there a reason why they are called A1, A2 and B1, rather than A1, A2 and A3?

There is no particular reason, it was a minor mistake in .tex-file and it is now corrected.

**2 Comments and Author response - Referee 2**

**2.1 Main comments**

*There is a large difference between the EDI and CODIF perpendicular velocity data. The CODIF perpendicular velocities have similar values to CODIF parallel velocities. This velocities go up to 120 km/s, and are def- initely not from the convection. EDI data give values of around 15 km/s which is what we expect convection to be. At this point we do not know how to explain the*

*CODIF perpendicular velocity measured in the cusps.*

You do not comment on that in the new version. I think it should be mentioned at the very least. Why trusting CODIF Vpar if Vperp is believed to be wrong? Vpar from CODIF is one of the major measurements this work employs. This has to be addressed somehow. I am not familiar with the CODIF dataset, is it a well-known problem? Can you provide references that support this way of using the data?

The comment on Vperp is now added. It is not that we do not believe in Vperp, we do not know how to explain them, and as i can see from the literature, no one has addressed this issue. One thing we can say with certainty is that Vperp from CODIF is not the convection, when we rescale it to ionospheric cusp convection velocities compare it to the ionospheric cusp convection velocities the result is around few km/s which is far grater then what is measured in ionosphere. When we scale the EDI measurements to the ionosphere, they are in agreement with the ionospheric measurements.

*Another main concern to me is if the dataset you use corresponds truly to cusps observations. For EDI you use TS96 to decide if you are in the cusps or not only, right? You should check other parameters as well when available, as for instance plasma beta.For CODIF dataset you do a much more accurate filtering of your dataset.*

*The plasma beta number is not always available when we have EDI data. We have decided to analyse each dataset separately and than combine the average values to get our estimate.*

You need to comment on the possible implications (or alternatively justify) the different treatment of the 2 datasets. You combine measurements of different nature, filtered using different criteria. I am worried, as mentioned before, whether your EDI measurements truly correspond to cusps measurements most of the time or not. Could you please develop on that?

Regarding the EDI measurements, yes using the model is not the most accurate way of data selection, but we are certainly not in the solar wind (EDI does not work in solar wind). We may be in polar caps in some cases, but have in mind that we are working with grand averages. The convection velocity obtained form the cusps would not be very different from the one in the polar caps (unlike the parallel velocities which are strongly modulated by heating). Also, the convection velocities obtained from EDI seems sensible,

and are what we would expect in the cusps (100-1000 m/s in ionosphere). This explanation is now added into the manuscript.

**2.2   Detailed comments**

P1 L16 an important

It is now corrected.

P2 L21 extra '>' ?

It is now corrected.

P8 L7 'on the TS96...'

It is now corrected.

P13 L15 '0' should be part of the exponent. What is the effective cross-section of the tail you consider? What is the total average O+ flux (ions/s)?

The average flow cross section is now changed to the total average flux.

[revised manuscript text omitted]

* * *
[c4] dataset
[c5] till
[c6] number
[c7] number
[c8] *Text added.*
[c9] of
[c10] the
[c11] the
[c12] *Text added.*
[c13] *Text added.*
[c14] *Text added.*
[c15] *Text added.*
[c16] *Text added.*
[c17] also
[c18] these two populations with
[c19] *Text added.*
[c20] *Text added.*
[c21] *Text added.*
[c22] with
[c23] *Text added.*
[c24] *Text added.*
[c25] *Text added.*

ion velocity measured with CODIF and $\boldsymbol{b}$ is the direction of the magnetic field. We have then easily calculated the perpendicular velocity as: $\boldsymbol{v}_\perp = \boldsymbol{v_O} - v\|\boldsymbol{b}$. The perpendicular $O^+$ velocities are comparable to the parallel velocities. We believe that the $\boldsymbol{v}_\perp$ values are too high and cannot explain convection. As of now we do not know how to explain the perpendicular velocities and choose to ignore them in this study, and use more accurate EDI measurements instead.

In total we have 1422 hours of CODIF measurements. The distribution of CODIF measurements is shown in the left panel of figure 2. Here we can see the difference in data coverage between the two instruments (EDI and CODIF). The main reason for this asymmetry are the technical restrictions of the instruments. EDI has fewer measurements closer to the magnetopause because of [c1]the higher variability of magnetic field, while CODIF has more measurements closer to the magnetopause [c2]due to of higher fluxes in this region. In addition to EDI and CODIF Cluster data we also used solar wind dynamic pressure, Dst and IMF data from the OMNI dataset (King and Papitashvili, 2005). [c3]For IMF and solar wind dynamic pressure we used one minute values and for Dst we have implemented a simple linear interpolation in order to get the one minute values.

**3   Method**

The method used is a combination of the ones described in Haaland et al. (2012) and Li et al. (2012). If the out-flowing ions can be traced to closed magnetic field lines before they reach the distant X-line at ca $-100\ R_E$ (e.g., Grigorenko et al., 2009; Daly, 1986), we [c4]infer that they are captured and returned to the magnetosphere. If they reach the X-line before being convected to the plasma sheet, the ions will be lost into the solar wind. For the highest energies, some of the ions will escape into the dayside magnetosheath directly before being convected into the plasma mantle. [c5]One issue here is the position of the distant X-line, which is not permanent, but can vary with geomagnetic conditions. Since we do not know the exact location of the distant X-line in relation to the geomagnetic conditions, we have decided to use the fixed X-line and [c6]comment on its effect on the results in the discussion [c7]section. Another issue is the forming of the near [c8]Earth X-line (around $X_{GSM} = -20\ \mathrm{R_E}$) during active geomagnetic conditions. At this
* * *
[c1] *Text added.*
[c2]
[c3] *Text added.*
[c4]
[c5]
[c6]
[c7] *Text added.*
[c8] *Text added.*

[revised manuscript text omitted]

* * *
[c1] ; Black

[c2] *Text added.*

[c3] Roughly, if

[c4] *Text added.*

[c5] in less than

[c6] *Text added.*

**Table 2.** Estimated fate of oxygen ions expressed as percentages of outflow flux. $\Phi$ represents the flux, and subscripts $ms$, $sw$ and $ps$ represent magnetosheath, solar wind and plasmasheet respectively. $\sigma_{par}$ represent the standard deviation of the parallel initial velocities.

| | $\langle v_{par} \rangle$ | $\langle v_{par} \rangle - \sigma_{par}$ | $\langle v_{par} \rangle + \sigma_{par}$ |
|---|---|---|---|
| $\Phi_{ms}$ | 18 % | 15 % | 19 % |
| $\Phi_{sw}$ | 50 % | 37 % | 63 % |
| $\Phi_{ps}$ | 31 % | 48 % | 18 % |

In figure 9 we show the results of the tracing on the sampling bins (starting positions of the tracing) i.e. the colors indicate where the tracing will end starting from each bin. Colors used are the same as in figure 8. The average cusp ion outflow is [c7]$3.9\times 10^{24}$ s$^{-1}$ and the estimated percentages of oxygen flux which end up in each region [c8]are given in table 2.

From our estimation, on average 31 % of the total oxygen flux from the high altitude cusp [c5]will be convected to the plasma sheet. The further fate of these ions and transport inside the plasma sheet is beyond the scope of this paper, but it is reasonable to assume that a fraction of the recirculated ions are eventually lost through plasmoid ejections, through the magnetopause and other loss processes.

We also present the resulting oxygen outflow for different storm conditions, using the Dst index as a proxy for storm conditions. For quiet conditions we used positive Dst values, for moderate storm conditions we used Dst values between 0 and $-20$ nT, and for active storm conditions we used Dst values below $-20$ nT. For quiet and active storm conditions for nightside measurement bins ($X_{GSM} \leq -1$ R$_{\mathrm{E}}$) the coverage is rather poor, but this is not a major problem, since the oxygen fluxes are rather low under these conditions, thus not affecting the overall results significantly. The threshold for active storm conditions might seem a bit high, but for [c6]a lower threshold we have [c7]a much smaller dataset and a lot more [c8]data gaps [c9](see appendix figures A1, A2, A3). The results of [c10]the tracing for different storm conditions are given in Figure 10. As seen from this figure the results are highly dependent on
* * *
[c7]

[c8] *is*

[c5]

[c6] *Text added.*

[c7] *Text added.*

[c8] *Text added.*

[c9] *Text added.*

[c10] *Text added.*

[Figure]

**Figure 8.** Tracing results using initial parallel velocities. Individual lines show the paths of particles from each measurement bin. Colors indicate the fate of oxygen ion: Blue indicate that they will return back to magnetosphere (mostly plasma sheet), red color indicate ions ending up in the solar wind; black indicate ions escaping into the dayside magnetosheath. Different panels represent cases for different starting velocities: The top panel shows results using average velocities, middle panel shows results using lower standard deviation velocities and the bottom panel shows results using upper standard deviation velocities.

[Figure]

**Figure 9.** [c1]Results of tracing for each starting bin. [c2]The different panels [c3]show various starting parallel velocities. Cases for the starting parallel velocities form left to right are: average parallel velocity, lower standard deviation parallel velocity and upper standard deviation parallel velocity.[c4]Transparent colours represent the bins with less than 30 minutes of the EDI data.

storm conditions. The gaps for active and quiet conditions would probably favour capture, since for moderate storm conditions this regions bins all show capture, but it would only change results by a couple percent [c11], due to the lower fluxes in those bins (See figure 7[c12]). [c13]The most interesting case is the tracing during active storm conditions, because most of [c14]the outflowing oxygen flux gets convected into [c15]the plasma sheet. During strong storms, both parallel [c16]velocities and convection velocities increase, but the increase in convection is stronger, causing a larger flux of oxygen ions into plasma sheet. The outflowing $O^+$ ions are deposited closer to Earth, for storm geomagnetic conditions. In figure 11 we show the results of the tracing in starting bins in the same way as in figure 9, but for various
* * *
[c11] *Text added.*
[c12] *Text added.*
[c13] *Text added.*
[c14]
[c15] *Text added.*
[c16] *Text added.*

[Figure]

**Figure 10.** The tracing results using parallel initial velocities for different storm conditions. [c1]The upper panel shows quiet conditions, middle panel shows moderate storm conditions, [c2]and the lower plot shows active storm conditions. The [c3]colors are the same as in fig. 8

geomagnetic conditions. The estimated percentages of fate of oxygen flux for various Dst conditions are given in table 3.

[Figure]

**Figure 11.** Results of tracing for each starting bin. Different panels depict various starting geomagnetic conditions. Cases for the starting parallel velocities form left to right are: quiet condition, moderate condition and active geomagnetic condition. The [c4]colors are the same as in figure 8

**Table 3.** Estimated fate of oxygen ions expressed as percentages of outflow flux. $\Phi$ represents the flux, and subscripts $ms$, $sw$ and $ps$ represent magnetosheath, solar wind and plasmasheet, assuming that the plasmasheet is limited by distant X-line at $X_{GSE} = -100$ $R_E$

|  | $Dst > 0$ nT | $0 > Dst > -20$ nT | $Dst < -20$ nT |
|---|---|---|---|
| $\Phi_{ms}$ | 9 % | 20 % | 12 % |
| $\Phi_{sw}$ | 62 % | 50 % | 15 % |
| $\Phi_{ps}$ | 29 % | 30 % | 73 % |

**5   Discussion**

In terms of oxygen outflow escape from the high altitude cusps and plasma mantle regions we find that most of the oxygen escape the magnetosphere, as shown by Slapak et al. (2017). As pointed out by Seki et al. (2002) and Ebihara et al. (2006), oxygen ions with low energies ($< 1$ keV) will end up in near tail plasma sheet or in ring current. Our results show that oxygen ions reaching the high altitude cusps will mostly escape the magnetosphere. On average, 50% of the oxygen outflow flux will end up in the solar wind beyond [c1]the distant X-line. 19% will escape directly into [c2]the dayside magnetosheath. This sums up to a total escape rate of 69 % of high altitude cusp oxygen flux. The rest, 31 % of the high altitude cusp flux is being convected [c3]to plasma sheet, mostly [c4]to the distant tail ($> 50$ $R_E$), as shown by the figure 8.

Another important issue is the escape-versus-capture ratio for different storm conditions. During quiet magnetospheric conditions, oxygen outflow and energization is relatively low, resulting in lower fluxes of oxygen in the high altitude cusp. However, in such cases, the magnetospheric convection is also low and consequently almost all of the outflowing oxygen escape. It is worth mentioning that in such cases IMF is mostly northward and can lead to lobe reconnection, resulting in sunward flow. This process can decelerate oxygen ions, and lead to their capture. [c5]Positive Dst periods are also characterised [c6]by sudden high $P_{DYN}$ [c7]changes (e.g., Boudouridis et al., 2007; Gillies et al., 2012). We did not [c8]did not differentiate sudden commencement form quiet conditions, since we used the average values [c9]of large dataset. We assume that such [c10]changes are increasing our average convection velocities and the oxygen outflow (which are low), and without them the results would not change much all together. During moderate storm conditions, results are similar to average conditions. For active storm conditions, the oxygen ion flux is high, and both the parallel velocity of the oxygen ions and the convection is higher. This leads to increase in both dayside magnetosheath escape and enhanced convection into the plasma sheet. Oxygen ions are more likely to escape into the dayside magnetosheath due to their high parallel velocities. Oxygen ions that get convected from the cusps into the plasma mantle will eventually be convected into the plasma sheet. There are also other processes which can further
* * *
[c1] *Text added.*
[c2] *Text added.*
[c3]
[c4]
[c5]
[c6]
[c7]
[c8]
[c9] *Text added.*
[c10]

[revised manuscript text omitted]